# A co-transcriptional ribosome assembly checkpoint controls nascent large ribosomal subunit maturation

Zahra A. Sanghai, Rafal Piwowarczyk ⓘ , Arnaud Vanden Broeck ⓘ & Sebastian Klinge ⓘ ✉

During transcription of eukaryotic ribosomal DNA in the nucleolus, assembly checkpoints exist that guarantee the formation of stable precursors of small and large ribosomal subunits. While the formation of an early large subunit assembly checkpoint precedes the separation of small and large subunit maturation, its mechanism of action and function remain unknown. Here, we report the cryo-electron microscopy structure of the yeast co-transcriptional large ribosomal subunit assembly intermediate that serves as a checkpoint. The structure provides the mechanistic basis for how quality-control pathways are established through co-transcriptional ribosome assembly factors, that structurally interrogate, remodel and, together with ribosomal proteins, cooperatively stabilize correctly folded pre-ribosomal RNA. Our findings thus provide a molecular explanation for quality control during eukaryotic ribosome assembly in the nucleolus.

Eukaryotic ribosome assembly involves the coordinated activity of more than 200 assembly factors that assist in the formation of small and large ribosomal subunits[1]. During the transcription of ribosomal DNA (rDNA) in *S. cerevisiae*, a 35S precursor transcript is generated in which the small subunit rRNA (18S) and two of the three large subunit (LSU) rRNAs (5.8S and 25S) are flanked by external transcribed spacers (5′ ETS and 3′ ETS) and are interspersed by internal transcribed spacers (ITS1 and ITS2). A unique and poorly understood challenge during co-transcriptional stages of ribosome biogenesis in the nucleolus is the correct assembly and stabilization of the 5′ ends of nascent pre-ribosomes through co-transcriptional ribosome assembly factors.

Miller spreads have highlighted the co-transcriptional appearance of assembly intermediates as terminal structures that first form for the small subunit, where the synthesis of a 5′ ETS ribonucleoprotein (RNP) precedes the assembly of the small-subunit processome[2–5]. The co-transcriptional formation of an uncharacterized LSU assembly intermediate around the 5′ end of the LSU pre-ribosomal RNA (pre-rRNA) is then followed by pre-rRNA cleavage at site A2 within ITS1, which separates small-subunit and LSU maturation[3].

However, how the structural integrity of the 5′ segment of the nascent LSU precursor is interrogated has so far remained elusive.

It is unclear how ribosome assembly factors implicated in this event can perform key checkpoint functions that require the presence and recognition of distinct pre-rRNA elements to facilitate the separation of small-subunit and LSU biogenesis.

The Noc1–Noc2 complex, together with Rrp5, has been implicated in mediating some of these functions as well as co-transcriptional cleavage at site A2, because Rrp5 is involved in both small-subunit and LSU assembly[6–11]. Chemical-biology approaches have highlighted that the Noc1–Noc2 complex and Rrp5 can stably bind to pre-ribosomal RNA mimics that contain a 5′ segment of the LSU[12,13]. However, the mode by which this complex may establish a checkpoint that can control the coordinated progression of both small-subunit and LSU assembly has so far remained unclear[14].

## Results

### Cryo-electron microscopy structure of the Noc1–Noc2 RNP

To characterize the mechanisms by which nascent LSUs are chaperoned in a co-transcriptional manner, we employed a system wherein pre-rRNA mimics are generated from a plasmid in vivo[4,13,15]. This system allowed for the synthesis of a pre-rRNA mimic starting at site A2 in ITS1 and terminating after domain VI of the 25S rRNA, followed by a set of

Laboratory of Protein and Nucleic Acid Chemistry, The Rockefeller University, New York, New York, USA. ✉e-mail: klinge@rockefeller.edu

MS2 RNA aptamers, as previously described[13] (Extended Data Fig. 1a). The expression of this pre-rRNA mimic resulted in the formation of mature LSUs, showing that the pre-rRNA mimic undergoes the entire LSU-assembly pathway (Extended Data Fig. 1b,c).

By using the pre-rRNA mimic with MS2 RNA aptamers and tagged Noc1 as baits for affinity purification, we were able to isolate an early co-transcriptional assembly intermediate (hereafter referred to as the Noc1–Noc2 RNP) (Extended Data Fig. 1). Similarly, by using the same pre-rRNA mimic with MS2 RNA aptamers and tagged Noc2, we were able to isolate the late nucleolar pre-60S assembly intermediate (state E, containing Noc2–Noc3)[16], indicating that our pre-rRNA mimic proceeds along a physiologically relevant assembly pathway (Extended Data Fig. 2). To determine the structure of the Noc1–Noc2 RNP, extensive 3D classification was performed on eight cryo-electron microscopy (cryo-EM) datasets to first obtain a well-defined module containing domain II of the 25S rRNA, before resolving a more flexible domain I and 5.8S rRNA module (Extended Data Figs. 1e–g and 3–5, Table 1).

The resulting composite cryo-EM reconstruction contains domains I and II of the 25S rRNA, the 5.8S rRNA and ITS2. This arrangement of domains is expected for the earliest co-transcriptional ribosome-assembly intermediate visualized in Miller spreads, as the protein composition of subsequent intermediates changes significantly only once domain VI has been transcribed and assembled for post-transcriptional maturation[12,13] (Fig. 1a,b). The overall structure of this complex highlights that the domain I and II modules act as largely independent units that, in contrast to the subsequent assembly states, are not yet joined through ribosomal proteins and rRNA[16–18]. In agreement with prior biochemical data, the 5.8S rRNA is largely flexible, with the most ordered regions corresponding to segments that form base pairs with the 25S rRNA[19].

## The Noc1–Noc2 complex remodels root helices of the large subunit

The most prominent feature of this particle is the Noc1–Noc2 complex, which is responsible for remodeling all root helices formed by the 5.8S rRNA and domains I and II of the 25S rRNA (helices 2, 25 and 26) (Fig. 1c,d and Extended Data Fig. 6). Because root helices form the base of each of the 25S rRNA subdomains (I–VI), and hence the architectural core of the LSU[20,21], their early recognition by the Noc1–Noc2 complex indicates a key point of quality control. The Noc1–Noc2 complex not only binds to several critical root helices, but specifically encircles helix 2 through molecular shape complementarity in a non-sequence-specific manner (Fig. 1d). Helix 2 is a crucial root helix of the forming LSU: it contains the 5′ end of the LSU pre-rRNA, and initiates the formation of the polypeptide exit tunnel, one of the essential functional centers of the LSU, which is formed in a tightly controlled manner during subsequent assembly stages in the nucleolus[16–18]. Although the Noc1–Noc2 complex is evolutionarily related to the Noc2–Noc3 and Nop14–Noc4 complexes[22], it is the only member of this family that can directly chaperone pre-ribosomal RNA, in this case helix 2 of the nascent LSU pre-rRNA (Extended Data Figs. 7 and 8). Further, we observed an unprocessed 5′ end of the LSU pre-rRNA that extends beyond the B1S cleavage site, which later generates the mature 5′ end of the 5.8S rRNA (Fig. 1d and Extended Data Fig. 6d).

## Binding of Noc1–Noc2 serves as a checkpoint in LSU assembly

The strategic position of Noc1–Noc2 within the Noc1–Noc2 RNP is key to its primary function during the establishment of a co-transcriptional ribosome-assembly checkpoint. In conjunction with assembly factors Mak16 and Rrp1, Noc1–Noc2 pre-assembles the highly intertwined rRNA domains I and II, thereby positioning these domains for co-operative stabilization by ribosomal proteins (Fig. 2a and Extended Data Fig. 9). Beyond the immediate formation of the co-transcriptional ribosome assembly checkpoint, the Noc1–Noc2 complex further controls post-transcriptional ribosome assembly at four levels: (1) RNA topology, (2) RNA processing, (3) the incorporation of ribosomal

**Table 1 | Cryo-EM data collection, refinement and validation statistics**

| | Noc1–Noc2 RNP (EMD-27919), (PDB 8E5T) | Noc2–Noc3 RNP (EMD-27910) |
|---|---|---|
| **Data collection and processing** | | |
| Magnification | ×22,500 | ×22,500 |
| Voltage (kV) | 300 | 300 |
| Electron exposure (e−/Å²) | 37.9 | 37.9 |
| Defocus range (μm) | −1.0 to −3.0 | −1.5 to −3.0 |
| Pixel size (Å) | 1.3 | 1.3 |
| Symmetry imposed | $C_1$ | $C_1$ |
| Initial particle images (no.) | 977,232 | 669,811 |
| Final particle images (no.) | 158,915 | 75,358 |
| Map resolution (Å) | 4.0 | 3.69 |
| FSC threshold | 0.143 | 0.143 |
| Map resolution range (Å) | 2.8–11.5 | 2.8–8.5 |
| **Refinement** | | |
| Initial model used (PDB code) | 6C0F | |
| Model resolution (Å) | 3.7 | |
| FSC threshold | 0.143 | |
| Model resolution range (Å) (FSC 0,0.143,0.5) Masked | 3.6/3.7/7.0 | |
| Map sharpening B factor (Å²) | −44.7 | |
| Model composition | | |
| Non-hydrogen atoms | 55,125 | |
| Protein residues | 5158 | |
| Nucleic acid | 968 | |
| Ligand | Zn-1, Mg-33 | |
| B factors (Å²) | | |
| Protein | 83.71/491.41/148.87 | |
| Ligand | 51.94/257.32/115.29 | |
| R.m.s. deviations | | |
| Bond lengths (Å) | 0.004 (0) | |
| Bond angles (°) | 0.927 (5) | |
| **Validation** | | |
| MolProbity score | 1.44 | |
| Clashscore | 3.96 | |
| Poor rotamers (%) | 0 | |
| Ramachandran plot | | |
| Favored (%) | 96.21 | |
| Allowed (%) | 3.69 | |
| Disallowed (%) | 0.10 | |

proteins and (4) the chronology of ribosome-assembly-factor association (Fig. 2).

First, at the level of RNA topology, the Noc1–Noc2 complex not only separates domains I and II for independent maturation, but also prevents the integration of domain VI, which is a post-transcriptional event (Figs. 1 and 2 and Extended Data Fig. 9)[16–18]. Key links between domains I and II, such as helix 19, are prevented from establishing contacts, and Noc1 substantially overlaps with domain VI, thereby preventing its incorporation during early co-transcriptional stages of LSU assembly (Fig. 2 and Extended Data Fig. 9).

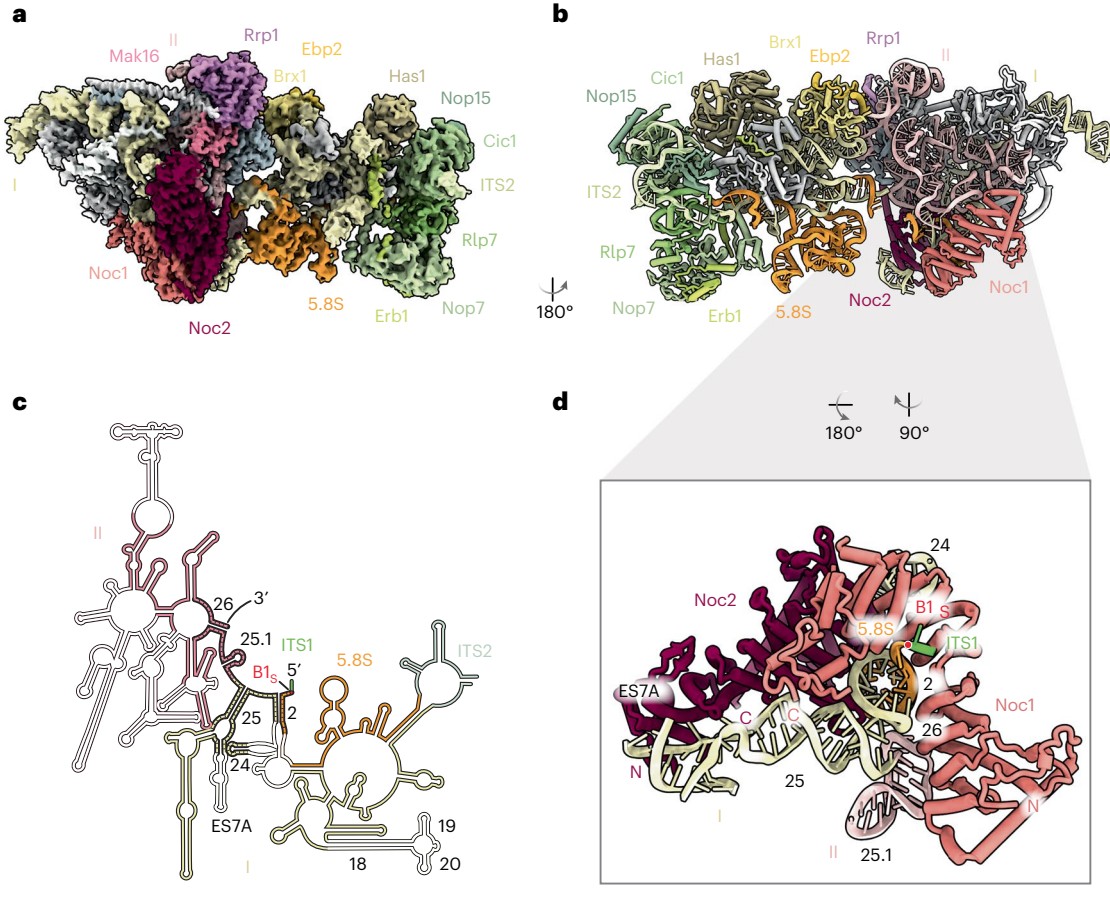

**Fig. 1 | Cryo-EM structure of the co-transcriptional Noc1–Noc2 RNP.**
**a**, Cryo-EM reconstruction of the Noc1–Noc2 RNP, visualizing domains I and II of the 25S rRNA with ITS2 and the 5.8S rRNA, and associated assembly factors and ribosomal proteins. **b**, Corresponding atomic model of the Noc1–Noc2 RNP, rotated by 180°. **c**, rRNA secondary structure diagram of the yeast LSU rRNA visualized in the Noc1–Noc2 RNP cryo-EM reconstruction. rRNA elements present in the structure are color-coded. Disordered segments are colored in white, and remodeled segments are indicated by dashed lines, with relevant helices numbered in black. **d**, The Noc1–Noc2 dimer clamps around root helix 2 of the pre-rRNA that is formed by base-pairing of the 5.8S and 25S rRNAs, and remodels root helices 25 and 26.

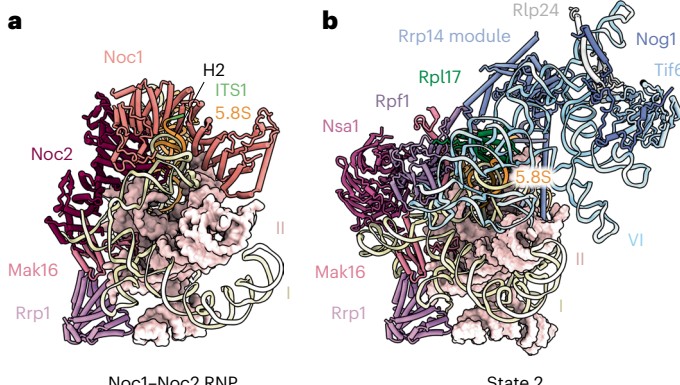

**Fig. 2 | The Noc1–Noc2 complex controls early LSU assembly. a**, Cartoon representation of color-coded assembly factors chaperoning domains I and II of the Noc1–Noc2 RNP. Domain II is shown in surface representation, and root helix 2 (H2) and ITS1 are indicated. **b**, A comparative view of state 2 (PDB: 6C0F, superimposed on Rrp1), with cartoon representation of color-coded assembly factors chaperoning domains I, II and VI. Domain II is shown in surface representation, and ribosomal protein Rpl17 is indicated in green.

Second, given the position of Noc1 with respect to the 5′ end of the 5.8S rRNA, Noc1 may also serve as a physical barrier preventing exonucleolytic trimming of ITS1 beyond site B1S (Figs. 1d and 2a).

Third, at the level of ribosomal protein incorporation, the structure rationalizes previous genetic and biochemical data showing that Rpl17 only gradually associates with early LSU assembly intermediates, as its binding site is initially occupied by Noc1 (ref. 23) (Fig. 2 and Extended Data Fig. 9c–f). Similarly, the Noc2 carboxy terminus interferes with the incorporation of Rpl26, as it prevents the ordering of helix 19 of the 25S rRNA, thereby eliminating a key binding site for Rpl26 at the junction of domains I and II. The importance of critical interfaces between domains I, II and the 5.8S rRNA is further highlighted by the finding that key stabilizers of these sites include Rpl17 (RPL17 in humans) and Diamond-Blackfan anemia proteins Rpl26 (RPL26) and Rpl35 (RPL35) (Extended Data Fig. 9c–f)[24,25].

Fourth, at the level of ribosome-assembly-factor chronology, the Noc1–Noc2 dimer is presumably recruited to early LSU assembly intermediates through an interaction with the Rrp5 amino terminus, which could not be assigned in our reconstruction. Although Noc2 contacts Mak16, it prevents the Mak16-associated ribosome-assembly-factor complex Rpf1–Nsa1 from binding near domains I and VI (Fig. 2).

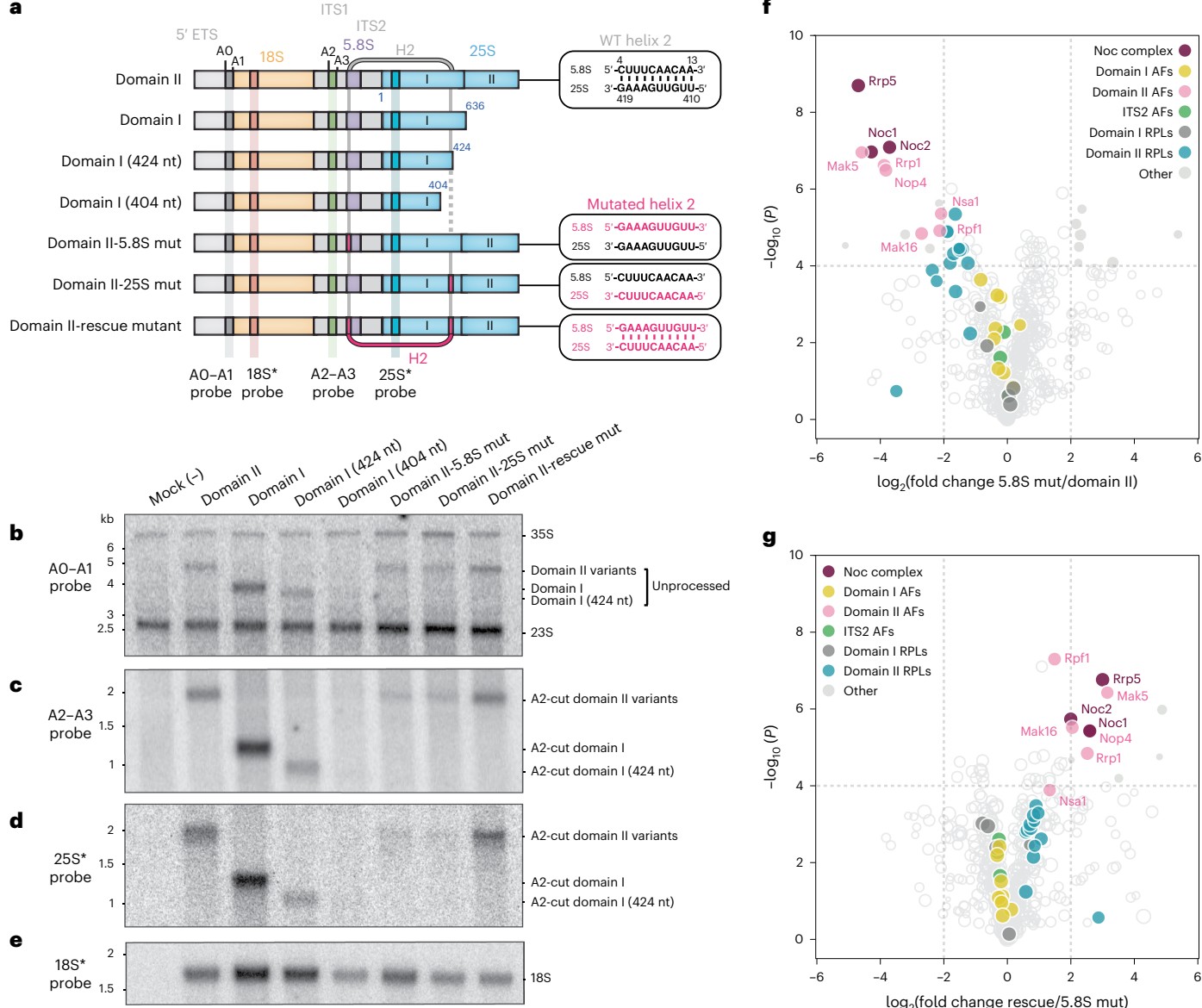

**Fig. 3 | Binding of Noc1–Noc2 verifies correct formation of helix 2.**
**a**, A schematic of rRNA mimics (domain II, domain I, helix 2 truncations and mutated constructs) expressed in yeast and analyzed by northern blot. Wild-type (WT) and mutated helix 2 (H2) are depicted in gray and magenta, respectively, with a dotted line representing the missing fragment of H2. **b**–**e**, Northern blot analysis of total RNA extracted from whole cells transformed with the LSU rRNA variants shown in **a**, probed with an A0–A1 probe (**b**), an A2–A3 probe (**c**), a 25S* (specifically inserted) probe (**d**) or an 18S* (specifically inserted) probe (**e**). Northern blot experiments were repeated independently (*n* = 2), with

similar results. **f**,**g**, Comparative mass spectrometry analysis of proteins (RPLs, ribosomal proteins; AFs, assembly factors) associated with LSU rRNA variants starting at site A2 and terminating with domain II. **f**, Comparison of proteins associated with domain II or mutated domain II-5.8S. **g**, Comparison of proteins associated with domain II-rescue mut and domain II-5.8S mut. log$_2$(fold change in protein abundance) is expressed as the mean of label-free quantification values of three replicates. *P* values were determined by a two-tailed Student's *t*-test, and a *P* value of 0.0001 was chosen as the cut-off for statistical significance.

In addition, Noc1 sterically clashes with the Ssf1–Rrp15 heterodimer, which stabilizes the junction between domains I and VI in the later state 2 of the nucleolar pre-60S[18].

Thus, by various modes of action, including steric hindrance and pre-rRNA remodeling, Noc1–Noc2 serves as a central nexus that stabilizes a co-transcriptional pre-rRNA folding intermediate while delaying subsequent post-transcriptional maturation events until transcription of the entire LSU pre-rRNA is completed.

## Helix 2 formation is interrogated by the Noc1–Noc2 complex
Previous biochemical data have shown that the transcription of the LSU pre-rRNA precedes co-transcriptional cleavage at site A2, suggesting

the presence of a ribosome assembly checkpoint[3]. In accordance with this data, the structure of the Noc1–Noc2 RNP indicates that the assembly of this particle likely serves as a checkpoint because the Noc1–Noc2 dimer interrogates the formation of root helices that form the core of the nascent LSU, encapsulates helix 2 and stabilizes the 5′ end of the 5.8S rRNA (Figs. 1 and 2a). To probe this ribosome assembly checkpoint, we designed assays to interrogate pre-rRNA processing in vivo and protein composition of different pre-rRNP species (Fig. 3). To investigate the ribosome assembly checkpoint in vivo, we recapitulated co-transcriptional ribosome assembly by transforming yeast strains with plasmids containing unique rDNAs that differ in their ability to form helix 2 (Fig. 3a). Since helix 2 is bound directly by Noc1–Noc2,

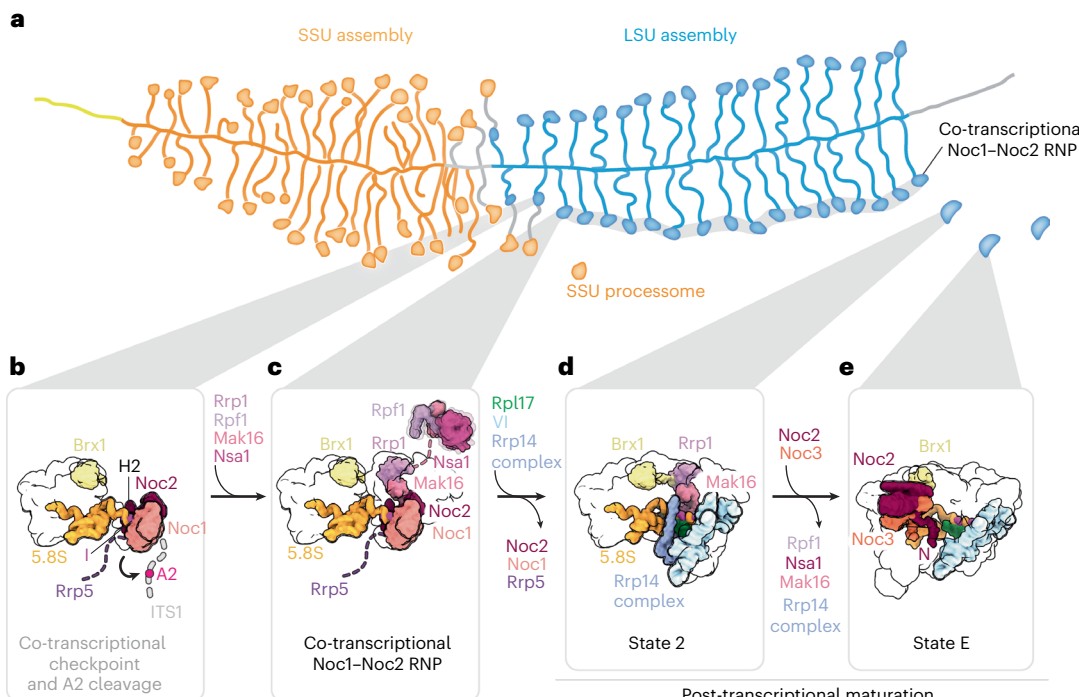

**Fig. 4 | A model for quality control during nucleolar LSU assembly.**
**a**, Representation of a Miller spread in which transcription of rDNA results in nascent pre-rRNA decorated by protein factors that chaperone the pre-rRNA into folded structures or terminal knobs. **b**, Schematic of how, upon transcription of domain I of the LSU, co-transcriptional assembly occurs with Noc1–Noc2–Rrp5 chaperoning the rRNA around helix 2, mediating a critical checkpoint and regulating the cleavage of the rRNA at site A2. **c**, Upon passing this quality check, the Noc1–Noc2 RNP is formed with domains I and II of the 25S rRNA incorporated (PDB 8E5T). **d**, Post-transcriptional maturation continues with state 2 (PDB 6C0F) as more ribosomal proteins join and further compaction of rRNA domains occur[18]. **e**, The Noc2–Noc3 RNP (state E) represents the later stages of nucleolar maturation, with most of the 25S rRNA domains ordered (EMD-27910).

we hypothesized that an inability to form a base-paired helix 2 should affect both pre-rRNA processing and the association of early assembly factors, such as the Noc1–Noc2 complex. To reflect endogenous co-transcriptional ribosome assembly, the designed plasmids encoded an entire small-subunit precursor (5′ ETS and 18S rRNA), followed by ITS1 and variants of LSU RNA elements that can be distinguished from the endogenous ribosomal subunits by virtue of distinct probes for northern blotting (Fig. 3a).

The structural integrity of helix 2 is key to this ribosome assembly checkpoint, and pre-rRNAs that included both strands of helix 2 (domain II, domain I and domain I truncated to 424 nucleotides (domain I (424 nt))) were detected as unprocessed transcripts (Fig. 3b) and could be further processed at site A2 to give rise to stable LSU precursors (Fig. 3c,d). By contrast, pre-rRNAs in which helix 2 was disrupted, either by truncation to remove one strand of helix 2 (domain I (404 nt)) or mutation of either strand of helix 2 (mutated domain II-5.8S (domain II-5.8S mut), mutated domain II-25S (domain II-25S mut)) were less stable (Fig. 3c,d). Strikingly, compensatory mutations that restore the RNA duplex within helix 2 (domain II-rescue mut, Fig. 3a) restore a stable precursor (Fig. 3b) that is effectively processed at site A2 (Fig. 3c–e).

To investigate whether the structural integrity of the RNA duplex of helix 2 is important for co-operative stabilization by proximal assembly factors, we performed comparative mass spectrometry with designed LSU rRNA precursors (Fig. 3f,g). Here, mutation of a single strand of helix 2 in the 5.8S rRNA (domain II-5.8S mut) selectively reduced the levels of proteins associated with domain II. The most affected proteins were those directly associated with helix 2 (Noc1, Noc2 and Rrp5), followed by proximal assembly factors associated with Mak16 (Rpf1, Rrp1 and Nsa1) and domain-II-associated ribosomal proteins, whereas ribosomal proteins associated with domain I were unaffected (Fig. 3f and Extended Data Fig. 10). By contrast, compensatory mutations within

helix 2 that restore the RNA duplex (domain II-rescue mut) selectively re-established binding of direct interactors of helix 2 (Noc1, Noc2 and Rrp5), as well as proximal assembly factors associated with Mak16 (Rpf1, Rrp1 and Nsa1) and domain-II-associated ribosomal proteins (Fig. 3g and Extended Data Fig. 10).

These data strongly support a model in which the formation of the nascent Noc1–Noc2 RNP constitutes an assembly checkpoint where local stabilization of core root helices (helices 2, 25 and 26) is communicated through the co-operative association of both assembly factors and ribosomal proteins. An inability to form a key root helix within the pre-rRNA (such as helix 2) prevents co-operative stabilization of the precursor by ribosomal proteins and assembly factors (such as Noc1–Noc2), thereby inhibiting downstream processing steps.

## Discussion

The combined structural and biochemical data presented here suggest that the coupled assembly of eukaryotic small and large ribosomal subunits can be described by the following model (Fig. 4): after the co-transcriptional assembly of the small-subunit processome (Fig. 4a), the transcription of ITS1, the 5.8S rRNA, ITS2 and most of the 25S rRNA domain I enables the formation of helix 2, which is sufficient to stabilize a co-transcriptional LSU assembly intermediate that can undergo cleavage at site A2 (Fig. 4b). The subsequent transcription of domain II generates the co-transcriptional assembly intermediate revealed in this study, which also contains the tethered assembly factor complex Mak16–Rpf1–Nsa1 and remains a part of the nascent pre-rRNP that is visualized in Miller spreads until rDNA transcription is completed (Fig. 4a,c). Following the complete transcription of the LSU pre-rRNA, post-transcriptional assembly events occur, where the departure of the Noc1–Noc2 complex, the integration of domain VI and the stable incorporation of Rpf1–Nsa1 lead to the formation of state 2

of the nucleolar pre-60S, in which domains I and II are joined (Fig. 4d). This is then followed by the association of the Noc2–Noc3 complex, which probes the polypeptide exit tunnel in state E (Fig. 4e). These mechanisms form the basis for the highly controlled initial stages of both co- and early post-transcriptional assembly and processing of the eukaryotic LSU.

Our structural and functional data in yeast shed light on the earliest events during co-transcriptional large ribosomal subunit formation, which are highly conserved in eukaryotes[26]. Together, these data provide a framework that explains a co-transcriptional ribosome assembly checkpoint centered around the 5′ end of nascent LSU precursors in the context of Miller spreads.

## Online content

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

## Methods

### Cloning of the MS2-tagged 25S rRNA and the MS2-3c-GFP construct

The LSU rDNA, beginning at the A2 site in ITS1 and terminating after domain VI of 25S, was cloned from the rDNA locus of the *Saccharomyces cerevisiae* strain BY4741 (MATa his3Δ leu2Δ0 met15Δ0 ura3Δ0) (ATCC 201388) into a derivative of the pESC_URA vector (Agilent Technologies). The resulting rRNA mimic (A2-domain VI) was tagged with five MS2-aptamer stem loops at its 3′ end and cloned downstream of a gal1 promoter and upstream of a CYC terminator. An adapted MS2-coat protein fused to an N-terminal nuclear localization signal (NLS), a hemagglutinin (HA) tag and a C-terminal 3C-protease-cleavable GFP (NLS-HA-MS2-3C-GFP) was cloned into a modified pESC plasmid suitable for genome integration in yeast, under a gal10 promoter with G418 resistance.

### Expression and purification of pre-60S particles

The pESC plasmid containing the MS2-3C-GFP under the gal10 promoter was transformed into the *S. cerevisiae* BY4741 strain for integration by selecting for G418 resistance. The resulting yeast strain with MS2-3C-GFP integrated was then subjected to endogenous tagging of a selected assembly factor—Noc1 (Mak21), Noc2 or Cic1 with a C-terminal streptavidin-binding-peptide (sbp) tag—using ClonNAT (Nourseothricin) selection. The resulting strains were used for the purification of the Noc1–Noc2 RNP (MS2-3C-GFP; Noc1-sbp), state E (MS2-3C-GFP; Noc2-sbp) and pre-60S particles for comparative mass spectrometry (MS2-3C-GFP; Cic1-sbp). These strains were subsequently transformed with the pESC_URA plasmids coding for pre-60S rRNA mimics for transient expression.

Yeast cultures were grown in Ura-synthetic drop-out (SD) medium containing 2% galactose (wt/vol) at 30 °C for 16–18 hours, reaching saturation (optical density at 600 nm (OD) of 5–6). Cells were then collected by centrifugation at 3,000g for 10 minutes at 4 °C, resulting in a cell mass of 20–30 grams. The cell pellet was washed with ice-cold double-distilled water (ddH$_2$O) twice, followed by a wash with ddH$_2$O containing protease inhibitors (E-64, Pepstatin, PMSF). Washed cells were immediately flash frozen in liquid nitrogen and lysed by four cycles of cryogenic grinding using a Retsch Planetary Ball Mill PM100. The freshly ground yeast powder was resuspended by vortexing in buffer A (50 mM Tris-HCl, pH 7.6 (20 °C), 150 mM NaCl, 5 mM MgCl$_2$, 1 mM DTT, 0.1% Triton X-100, PMSF, pepstatin, E-64), followed by centrifugation at 4 °C, 40,000g for 30 minutes to remove the insoluble component. The supernatant was then incubated with anti-GFP nanobody beads (Chromotek) for 3 hours at 4 °C, with gentle agitation. The beads were washed three times in ice-cold buffer A and once in buffer B (50 mM Tris-HCl pH 7.6 (20 °C), 150 mM NaCl, 5 mM MgCl$_2$, 1 mM DTT) before the bound proteins were eluted with 3C-protease cleavage for 1 hour at 4 °C. The eluate in buffer B was then applied to NHS-sepharose beads (Sigma), coupled with streptavidin for 1 hour at 4 °C with agitation. The pre-60S-bound streptavidin beads were washed once with buffer B before release from beads using buffer B containing 5 mM D-biotin. The eluted sample typically had an absorbance at 260 nm ($A_{260}$) of 1.0 to 2.5 mAU (Nanodrop 2000, Thermo Scientific) (Extended Data Fig. 1). The quality of the sample was assessed by SDS–PAGE and negative-stain electron microscopy (EM), and the sample was used for preparing cryo-EM grids. Mass-spectrometry analysis of the sample allowed for identification of expected protein components in the purified complex.

### Cryo-EM sample and grid preparation

Eight separate purifications of the Noc1–Noc2 RNP were performed to prepare eight cryo-EM grids that were used to collect eight datasets (DS1–DS8). The eluate in buffer B (above) was supplemented with 0.1% Triton X-100 (final concentration) before grid preparation. Copper grids of 400 mesh with lacey carbon and an ultra-thin carbon support film were used (Ted Pella, product no. 01824). A volume of 3–4 µL of sample ($A_{260}$ of ~1.5) was applied onto glow-discharged grids, which were incubated for 30 seconds and plunged into liquid ethane using a Vitrobot Mark IV robot (FEI Company) (95% humidity, blot force of 2–4 and blot time 3.5–4 seconds). Cryo-EM grids for the two datasets collected for the Noc2–Noc3 particle were prepared in the same manner.

### Cryo-EM data collection and image processing

**Noc1–Noc2 RNP.** A total of 18,046 micrographs were obtained over eight data collections on a Titan Krios (Thermo Fisher), at 300 kV with a K2 Summit detector (Gatan). SerialEM (v. 3.8)[27] was employed for data acquisition using a defocus range of 1.0–3.0 µm with a pixel size of 1.3 Å. Super-resolution movies (pixel size 0.65 Å) with 32 frames were collected using a total dose of 8 electrons per pixel per second, with an exposure time of 8 seconds and a total dose of 37.9 electrons per Å$^2$.

Upon data collection, the movies were gain corrected, dose weighted, aligned and binned to a pixel size of 1.3 Å using RELION 3.0's implementation of a Motioncor2-like algorithm, and the contrast transfer function (CTF) was estimated using CTFFIND 4.1 (ref. 28) within RELION[29]. Corrected and aligned micrographs were subjected to automated particle picking by CrYOLO (version 1.7.6)[30], which resulted in a total of 1,878,650 particles from all 8 data sets. Particle coordinates were then imported into RELION for subsequent steps. Particles were extracted with a box size of 220 pixels (pixel size of 2.6 Å/px), and were subjected to two-dimensional (2D) classification separately for each individual data set. After 2D classification, bad classes were removed and selected particles from each data set were combined, resulting in a total particle stack of 977,232 particles. The particles were subjected to a round of Bayesian polishing in RELION, followed by global search 3D classification into five classes ($K = 5$) using an initial 3D model obtained from an ab initio reconstruction from DS1 in RELION, low-pass filtered to 40 Å. The best two classes from this 3D classification were selected and their particles were re-extracted with a box size 440 pixels (pixel size of 1.3 Å/px). A combined total of 556,013 particles were used for a consensus 3D refinement with a solvent mask containing the entire particle, resulting in an overall resolution of 6.9 Å.

Owing to the observed inherent flexibility of the particle between domains I and II of the 25S rRNA, the particles were subjected to particle subtraction in RELION to remove signal from each domain, which were processed separately, in order to improve the quality and resolution of the density. Upon subtraction of the signal using individual masks around each domain, three-dimensional (3D) refinement followed by a round of 3D classification without image alignment into eight classes was performed, first for domain II ($K = 8$). The best class was selected, and checked for duplicates, resulting in a final particle stack of 158,915 particles. The particles were then subjected to 3 rounds of CTF refinement and Bayesian polishing, resulting in a final reconstruction of domain II at an overall resolution of 3.6 Å. This particle stack was then reverted back to its original form, and the signal from domain II was removed, providing a starting point to obtain the domain I reconstruction. The particles were 3D classified without image alignment into eight classes ($K = 8$), and 3 of the most similar classes were selected and combined, resulting in a total of 49,406 particles. These particles were then further classified with alignment into five classes ($K = 5$), and a single class was chosen with 17,104 particles. After 3 rounds of CTF refinement and particle polishing in RELION, a final reconstruction of domain I at a resolution of 4.72 Å was obtained (Extended Data Fig. 2). The local resolution of the maps was calculated with blocres[31] within cryoSPARC (v. 3.3.1)[32] (Extended Data Fig. 3). All reported resolutions are based on the gold-standard Fourier shell correction (FSC) criterion of 0.143, and FSC curves were corrected using high-resolution noise substitution methods in RELION 3.0. A composite map for the entire complex was generated using phenix.combine_focused_maps[33]. The same program was used to generate composite half maps that were used to calculate the overall resolution of the reconstruction at 4.0 Å.

**Noc2–Noc3 RNP.** A total of 8,249 micrographs from rounds of 2 data collection was obtained, on the Titan Krios (Thermo) using the same grid preparation and collection parameters, as was described above for the Noc1 particle. Particles were picked using CrYOLO (version 1.7.6), resulting in 253,724 particles in one data set and 416,087 particles in the other. The extracted particles (box size 440 pixels) were subjected to a round of 2D classification and 3D classification with alignment ($K = 5$) separately, to obtain one good 3D class from each of the data sets, which were then combined to obtain a final particle stack of 75,358 particles. The particles were then 3D-refined and put through 2 rounds of CTF refinement and Bayesian polishing, resulting in a final reconstruction of the Noc2–Noc3 particle at 3.69-Å resolution. The local resolution of the maps was calculated with blocres within cryoSPARC (v. 3.3.1) (Extended Data Fig. 3). All reported resolutions are based on the gold-standard FSC criterion of 0.143, and FSC curves were corrected using high-resolution noise substitution methods in RELION 3.0.

### Model building and refinement
Using the structure of the *S. cerevisiae* early nucleolar pre-60S particle (PDB 6C0F)[18] as a reference, common assembly factors and ribosomal proteins were manually located and fitted into the density. New assembly factors Noc1 and Noc2 were initially built de novo while using models predicted by AlphaFold[34] subsequently as references, and remodeled segments of the 25S and 5.8S rRNA were also built de novo into the domain II density. Rigid-body docking was used for domain I factors and rRNA. Model building was performed with COOT[35]. An annotated list of individual protein IDs, reference models and corresponding maps used for building can be found in Supplementary Table 1. The final model was refined against the composite map in PHENIX with phenix.real_space_refine using secondary structure restraints for proteins and RNAs[33]. Refinement and model statistics can be found in Table 1. All map and model analyses and illustrations were made using UCSF ChimeraX (version 1.2.5)[36] and the PyMOL Molecular Graphics System (version 2.3.5, Schrödinger).

### RNA extraction and northern blotting
The rRNA mimics were purified as described above, and RNA was extracted from the final eluate with 1 mL TRIzol (Life Technologies), according to the manufacturer's instructions. The whole cell RNA was extracted by resuspending 0.2 g yeast cells in 200 μL of TRIzol, followed by lysis by bead beating for a total time of 10 minutes. The total volume of TRIzol was then brought to 1 mL, and the extraction continued according to the manufacturer's instructions. One microgram of isolated rRNA was separated on a denaturing 1.2% formaldehyde-agarose gel (SeaKem LE, Lonza). After staining the gel in 1× SYBR Green II (Lonza) ddH$_2$O solution (pH 7.5) for 30 minutes, RNA species were visualized with a Gel Doc EZ Imager (Bio-Rad) (Extended Data Fig. 1) and then transferred onto a cationized nylon membrane (Zeta-Probe GT, Bio-Rad) using downward capillary transfer. The RNA was cross-linked to the membrane for northern blot analysis by ultraviolet (UV) irradiation at 254 nm with a total exposure of 120 mJ/cm$^2$ in a UV Stratalinker 2400 (Stratagene). Cross-linked membranes were initially stained with methylene blue dye to visualize the quality of transfer and then were incubated with hybridization buffer (750 mM NaCl, 75 mM trisodium citrate, 1% (wt/vol) SDS, 10% (wt/vol) dextran sulfate, 25% (vol/vol) formamide) at 65 °C for 30 minutes before addition of γ-$^{32}$P-end-labeled DNA oligonucleotide probe.

Oligonucleotide probe sequences were as follows:
25S probe*: 5′-AGGTACACTCGAGAGCTTCA-3′
18S probe*: 5′-CGAGGATCGAGGCTTT-3′
5′ ETS (A0–A1) probe: 5′-CCCACCTATTCCCTCTTGC-3′
ITS1 (A2–A3) probe: 5′-TGTTACCTCTGGGCCCCGATTG-3′

The probe was hybridized for 1 hour at 65 °C and then overnight at 37 °C. Membranes were washed once with wash buffer 1 (300 mM NaCl, 30 mM trisodium citrate, 1% (wt/vol) SDS) and once with

wash buffer 2 (30 mM NaCl, 3 mM trisodium citrate, 1% (wt/vol) SDS) for 20 minutes each at 45 °C. Radioactive signal was detected by exposure of the washed membranes to a storage phosphor screen, which was scanned with a Typhoon 9400 variable-mode imager (GE Healthcare).

### Mass spectrometry and comparative data analysis
Purified RNP samples were dried and dissolved in 8 M urea/0.1 M ammonium bicarbonate/10 mM DTT. After reduction, cysteines were alkylated in 30 mM iodoacetamide (Sigma). Proteins were digested with LysC (Endoproteinase LysC, Wako Chemicals) in less than 4 M urea, followed by trypsinization (Trypsin Gold, Promega) in less than 2 M urea. Digestions were halted by adding TFA and digests were desalted[37] and analyzed by reversed phase nano-liquid chromatography–tandem mass spectrometry (nano-LC–MS/MS) using a Fusion Lumos (Thermo Scientific). Data were quantified and searched against the *S. cerevisiae* Uniprot protein database (2019) concatenated with the MS2 protein sequence and common contaminations. For the search and quantitation, MaxQuant v. 2.0.3.0 (ref. 38) was used. Oxidation of methionine and protein N-terminal acetylation were allowed as variable modifications, and all cysteines were treated as being carbamidomethylated. The 'match between runs' option was enabled, and false-discovery rates (FDR) for proteins and peptides were set to 1% and 2%, respectively.

Protein abundances were expressed as LFQ (label-free quantification) values. Data were analyzed using Perseus (v.1.6.10.50)[39]. In short, LFQ values were log$_2$-transformed, followed by a filtering requiring that a protein must be matched in all three replicates for at least one condition. A two-tailed Student's *t*-test was carried out to confirm statistical significance of data. *Q* values resulting from FDR correction are listed in the Source Data for Figure 3.

### Statistics and reproducibility
No statistical method was used to predetermine sample size. Misaligned, damaged or contaminating particles were excluded from final reconstructions during image processing. For comparative IP–MS/MS experiments, data were filtered such that a protein must be present in all three replicates for at least one condition. During refinement, the gold-standard approach was used to randomly assign particles to half-sets of data that were independently averaged and compared to obtain resolution estimates. Other experiments did not require randomization. During single particle analysis, particles were randomly assigned into half-sets, thus no blinding was applied.

### Reporting summary
Further information on research design is available in the Nature Portfolio Reporting Summary linked to this article.

## Data availability
The cryo-EM maps and atomic models have been deposited in the Electron Microscopy Data Bank (EMDB) and the Protein Data Bank (PDB): Noc1–Noc2 RNP (EMD-27919, PDB 8E5T) and Noc2–Noc3 RNP (EMD-27910). Raw unaligned multi-frame movies and aligned micrographs of the Noc1–Noc2 RNP have been deposited in the Electron Microscopy Public Image Archive (EMPIAR-11379). The starting model used for model building of the Noc1–Noc2 RNP is PDB 6C0F. Normalized log$_2$-transformed LFQ mass-spectrometry data are provided as Source Data Fig. 3. Materials are available from S. K. upon reasonable request under a material transfer agreement with the Rockefeller University. Source data are provided with this paper.

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

## Acknowledgements

We would like to thank M. Ebrahim, J. Sotiris and H. Ng at the Evelyn Gruss Lipper Cryo-Electron Microscopy Resource Center at The Rockefeller University (RRID:SCR_021146) for assistance with grid screening and data collection. Mass spectrometry data were generated by the Proteomics Resource Center at The Rockefeller University (RRID:SCR_017797) using instrumentation funded by the Sohn Conferences Foundation and the Loena M. and Harry B. Helmsley Charitable Trust, and we particularly thank H. Molina, S. Heissel and C. Peralta for assistance. We thank F. Stengel and C. Sailer for sharing cross-linking and mass spectrometry data ahead of publication, O. Buzovetsky for assistance with model building and members of the Klinge laboratory for critical reading of the manuscript. This work was supported by funding from the G. Harold and Leila Y. Mathers Foundation (MF-2104-01554) and the National Institutes of Health grant GM143181 to S.K.

## Author contributions

Z. A. S. and S. K. conceived the study, Z. A. S. performed RNA analysis, cryo-EM sample preparation, data collection, processing and model building. R. P. performed biochemical experiments and mass spectrometry analysis, A. V. B. performed cryo-EM processing and model building, and all authors wrote and edited the manuscript.

## Competing interests

The authors declare no competing interests.

## Additional information

**Extended data** is available for this paper at https://doi.org/10.1038/s41594-023-00947-3.

**Correspondence and requests for materials** should be addressed to Sebastian Klinge.

**Peer review information** *Nature Structural & Molecular Biology* thanks David Taylor, Alan Warren, and the other, anonymous, reviewer(s) for their contribution to the peer review of this work. Sara Osman and Dimitris Typas were the primary editors on this article and managed its editorial process and peer review in collaboration with the rest of the editorial team. Peer reviewer reports are available.

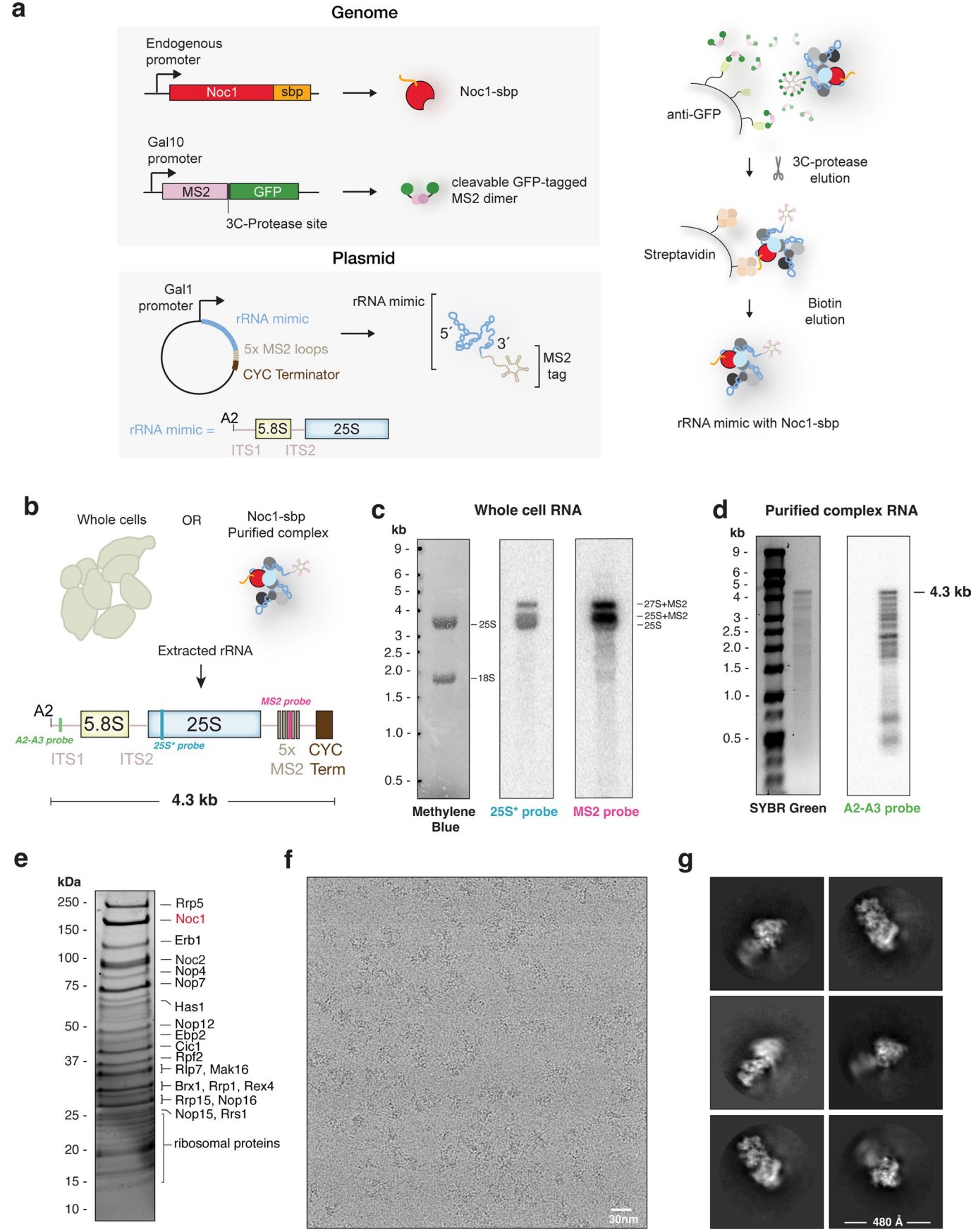

**Extended Data Fig. 1 | See next page for caption.**

**Extended Data Fig. 1 | Noc1-Noc2 RNP expression and purification scheme and cryo-EM images.** (**a**) Genomic integration and composition of the plasmid containing the rRNA mimic used for Noc1-Noc2 RNP expression in yeast, followed by schematic of the two-step purification of the Noc1-Noc2 RNP (**b**) Schematic of RNA extraction from whole cells (see panel c) and the purified Noc1-Noc2 RNP (see panel d). (**c**) Methylene blue-stained RNA gel and northern blot of RNA extracted from whole cells containing the Noc1-Noc2 RNP. (**d**) SYBR Green-stained RNA gel and northern blot of RNA extracted from purified Noc1-Noc2 RNP. (**e**) Representative SYPRO Ruby stained SDS-PAGE of purified fraction containing the Noc1-Noc2 RNP pre-ribosomal particles. Molecular weight markers are indicated on the left (MW) and bait protein (Noc1; red) and other bands were identified by LC-MS/MS analysis of eluate. RNA, protein gels and northern blot experiments (c-e) were repeated multiple times (n>3) with similar results. (**f**) Representative motion-corrected cryo-EM micrograph of the Noc1-Noc2 RNP particle sample. (**g**) Six most-populated 2D class averages (1.3 Å/px), extracted box size 440 pixels, mask diameter 480 Å.

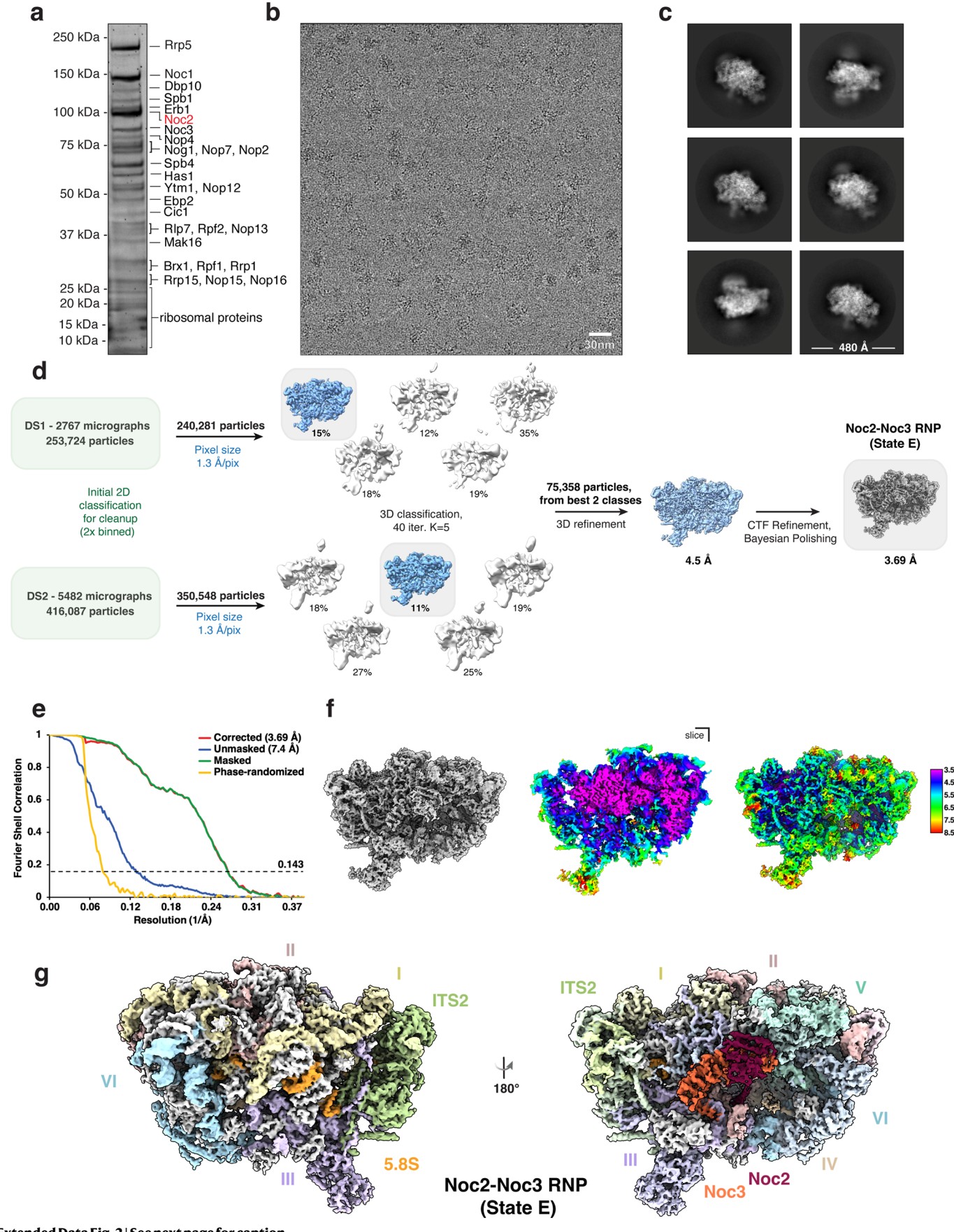

**Extended Data Fig. 2 | See next page for caption.**

**Extended Data Fig. 2 | Purification and structure determination of the Noc2-Noc3 RNP (State E). (a)** Representative SYPRO Ruby stained SDS-PAGE of purified fraction containing the Noc2-Noc3 RNP (State E) pre-ribosomal particles. Molecular weight markers are indicated on the left (MW) and bait protein (Noc2; red) and other bands were identified by LC-MS/MS analysis of the eluate. Protein gel (and purifications) were repeated multiple times (n>3) with similar results. **(b)** Representative motion-corrected cryo-EM micrograph of the Noc2-Noc3 RNP particle sample. **(c)** Six most populated 2D class averages (1.3 Å/px), box size 440 pixels, mask diameter 480 Å. **(d)** Cryo-EM data acquisition and processing workflow to obtain the final reconstruction of the Noc2-Noc3 RNP at a resolution of 3.69 Å. **(e)** Solvent-corrected FSC curves for the Noc2-Noc3 RNP, calculated in Relion 3.0, with reported resolutions determined at FSC=0.143 upon post-processing. **(f)** Local resolution estimation displayed in rainbow color on the cryo-EM map of the Noc2-Noc3 RNP. **(g)** Final cryo-EM reconstruction of the Noc2-Noc3 RNP with rRNA domains, assembly factors and ribosomal proteins (in gray) identified and colored.

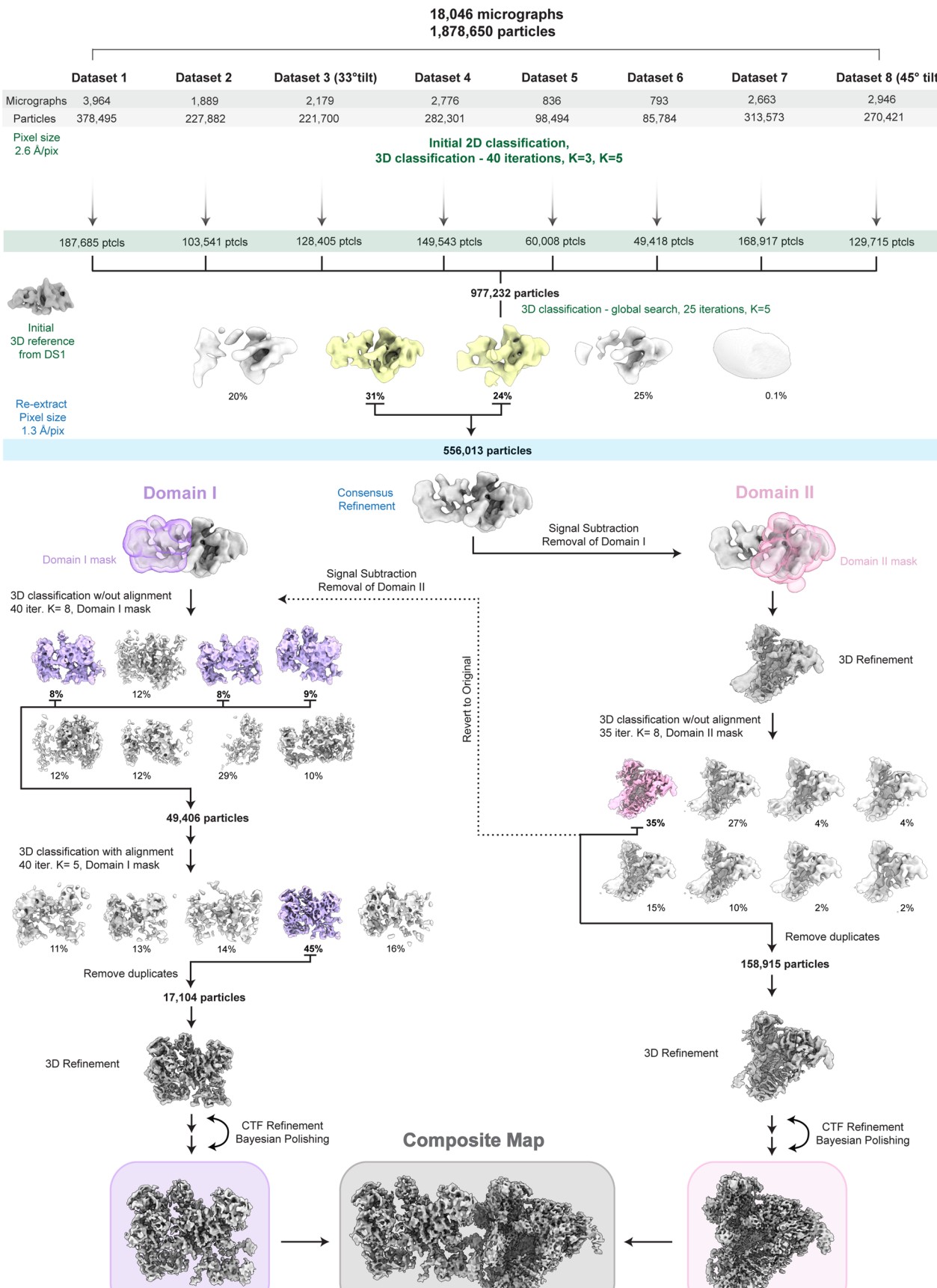

**Extended Data Fig. 3 | Data collection and structure determination of the Noc1-Noc2 RNP.** Cryo-EM data acquisition and processing workflow for the eight data sets collected to obtain the final reconstruction of the Noc1-Noc2 RNP.

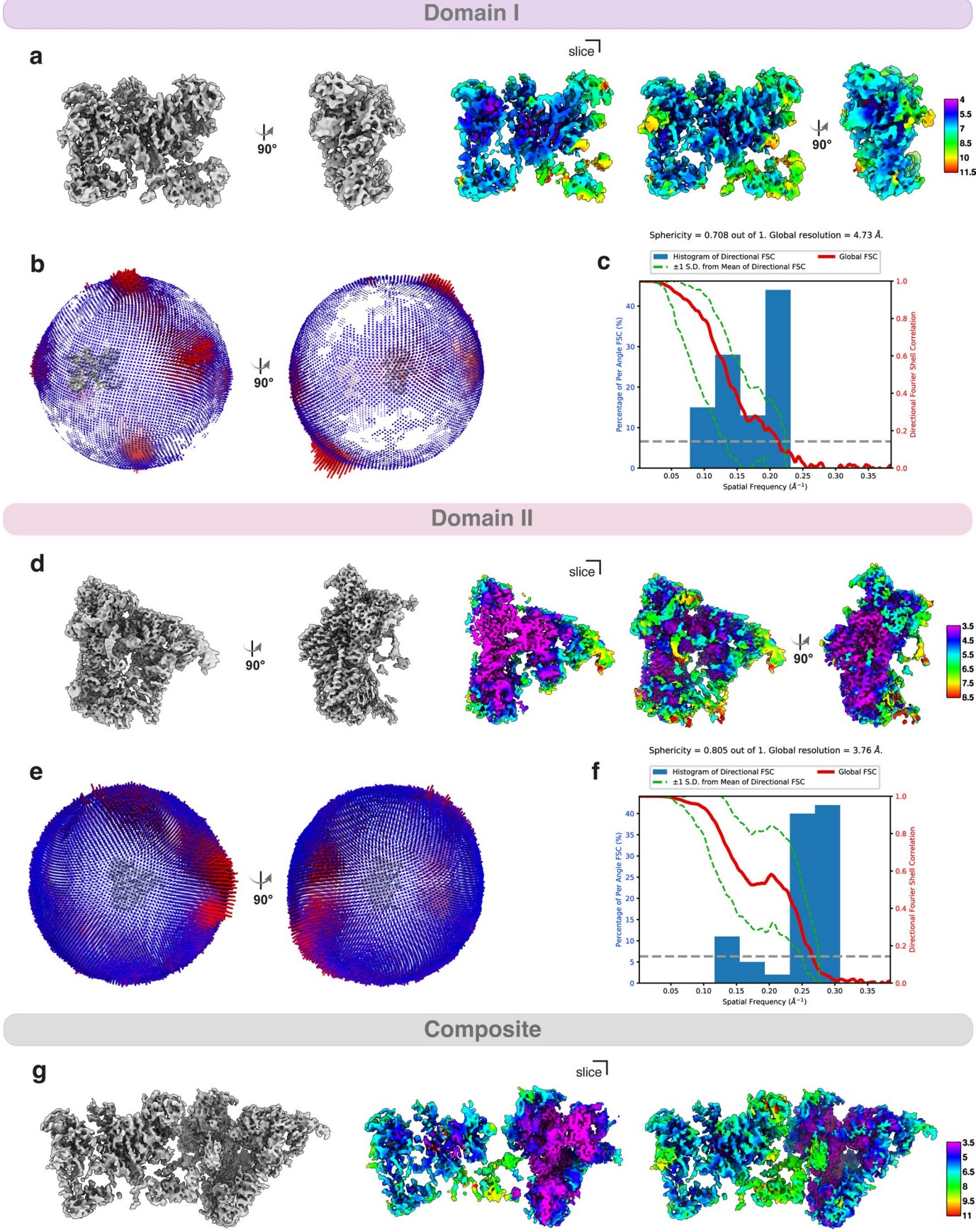

**Extended Data Fig. 4 | Local resolution estimation and 3DFSC curves of the Noc1-Noc2 RNP.** Local resolution estimation displayed in rainbow color on the cryo-EM map of **(a)** domain I, **(d)** domain II, **(g)** composite map shown in 2 different views as calculated by cryoSPARC. Euler angular distribution of the particles from the final refinement of **(b)** domain I, **(e)** domain II. 3DFSC curve for **(c)** domain I and **(f)** domain II reconstructions calculated by cryoSPARC.

## Domain I

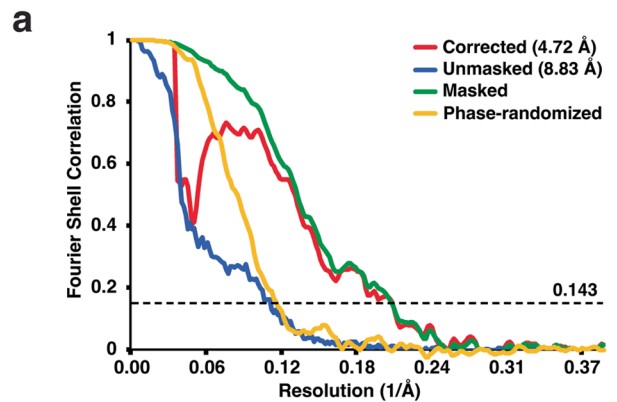

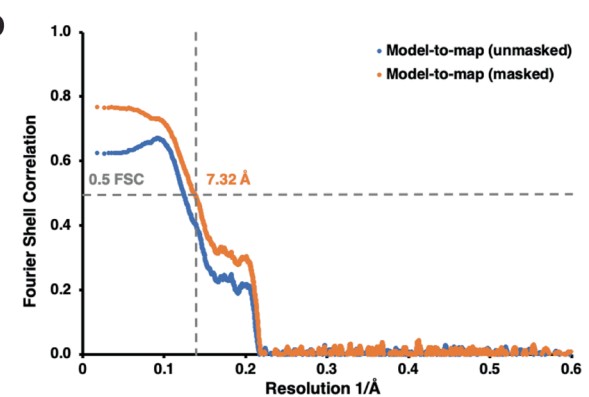

## Domain II

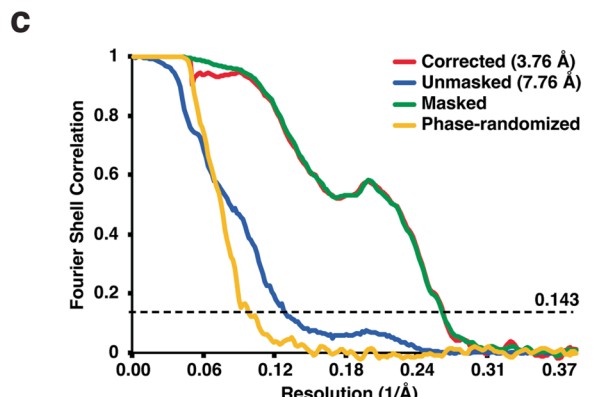

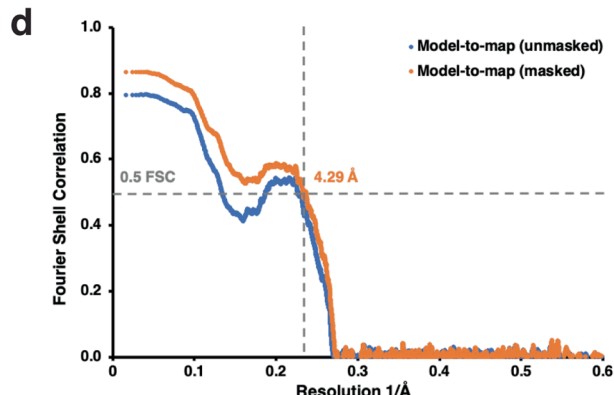

## Composite

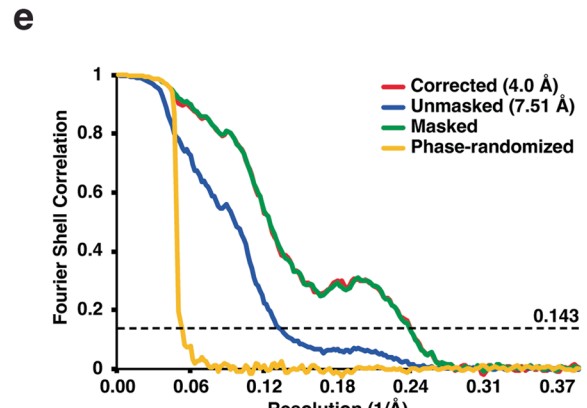

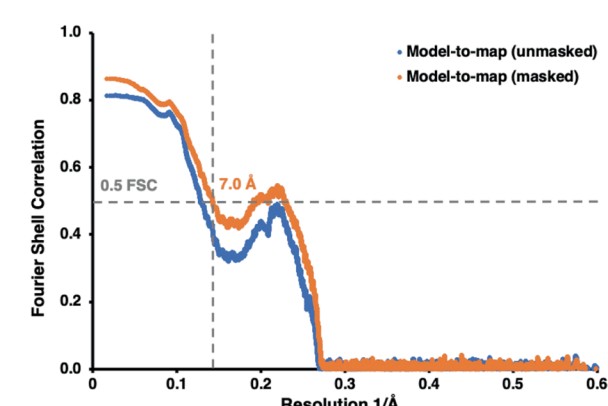

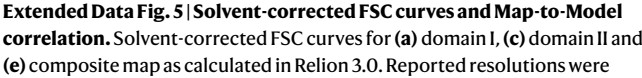

**Extended Data Fig. 5 | Solvent-corrected FSC curves and Map-to-Model correlation.** Solvent-corrected FSC curves for **(a)** domain I, **(c)** domain II and **(e)** composite map as calculated in Relion 3.0. Reported resolutions were determined at FSC = 0.143 upon post-processing in Relion 3.0. Map-to-model correlation between **(b)** domain I map and model, **(d)** domain II map and model and **(f)** composite map and model with resolutions reported at FSC = 0.5.

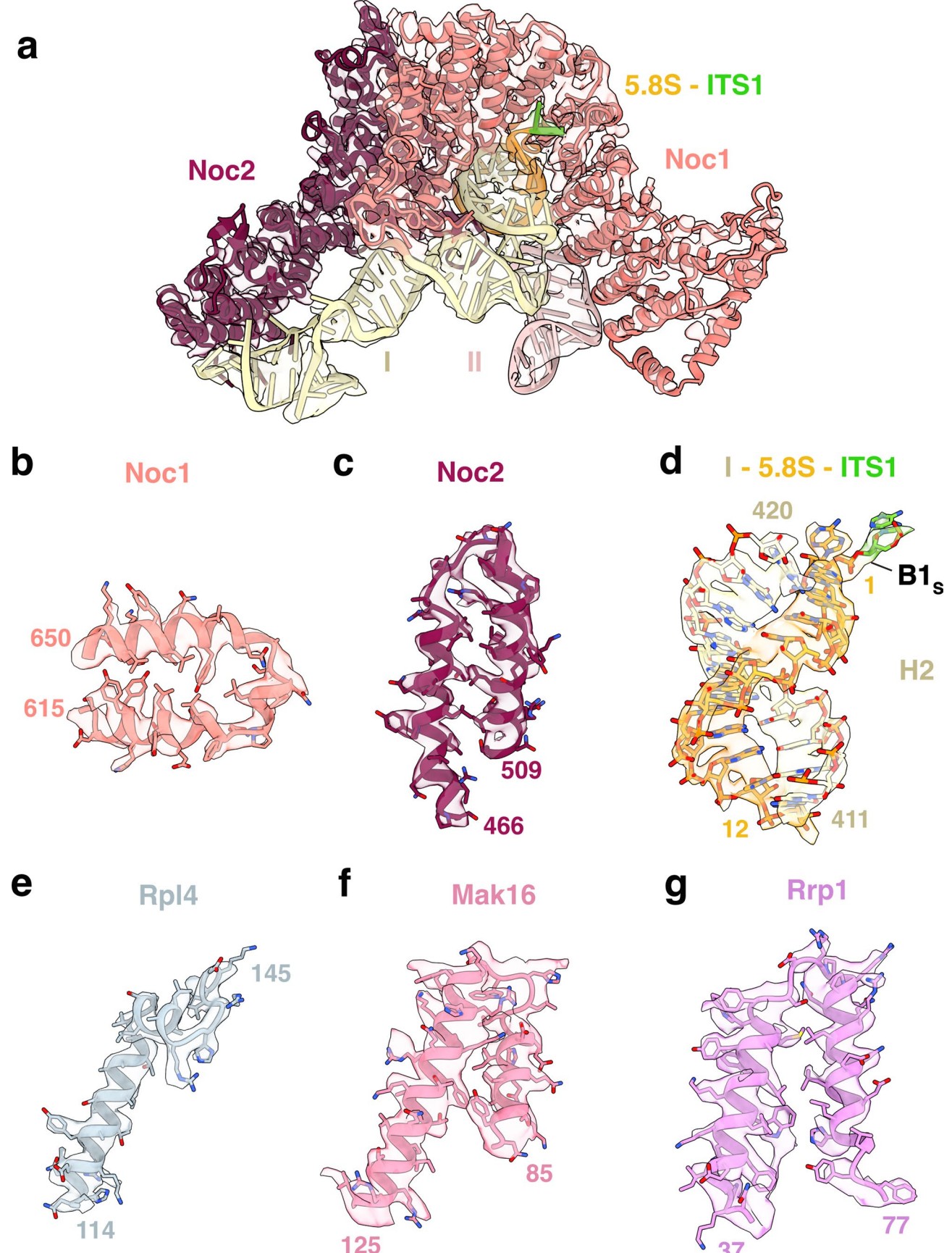

**Extended Data Fig. 6 | Representative cryo-EM densities and models for the Noc1-Noc2 RNP. (a–g)** Selections of representative densities of RNA and proteins of the Noc1-Noc2 RNP comprised of an overview of the Noc1-Noc2 proteins interacting with the rRNA of the 25S (domains I-II) and 5.8S (a), assembly factors (b,c,e,f), 25S and 5.8S rRNA from helix 2 (d), and ribosomal proteins (e). Densities are illustrated as continuous transparent volumes and models are shown as ribbons and sticks.

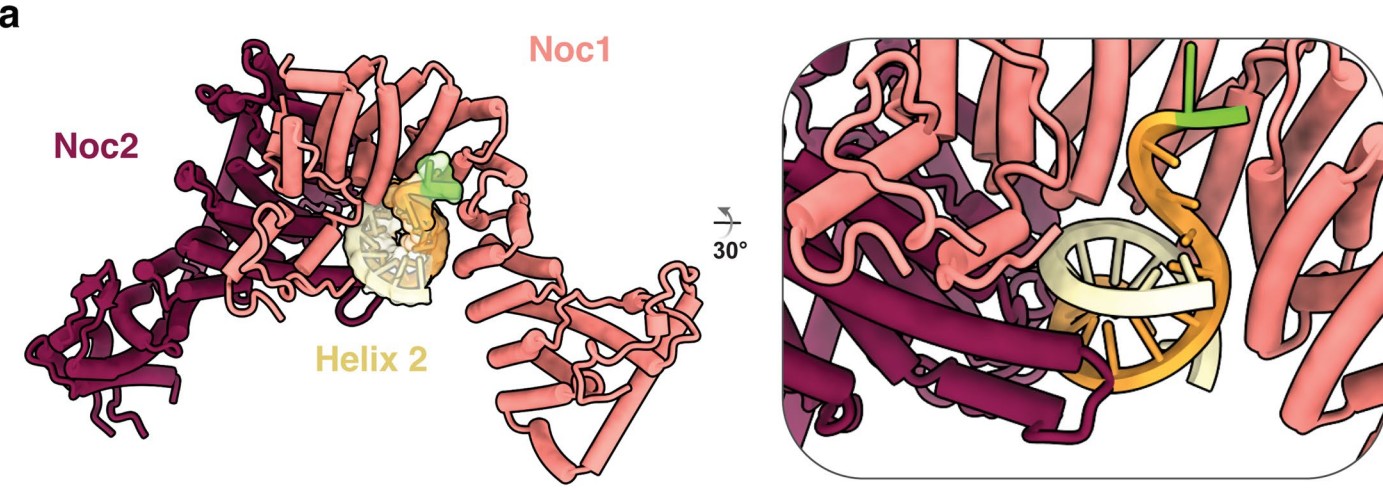

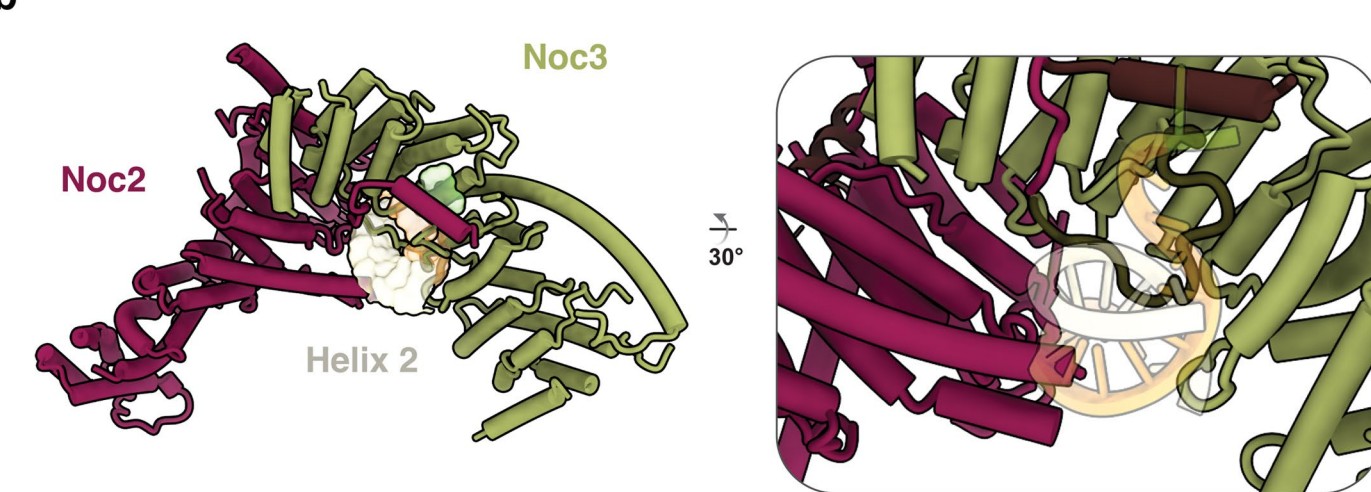

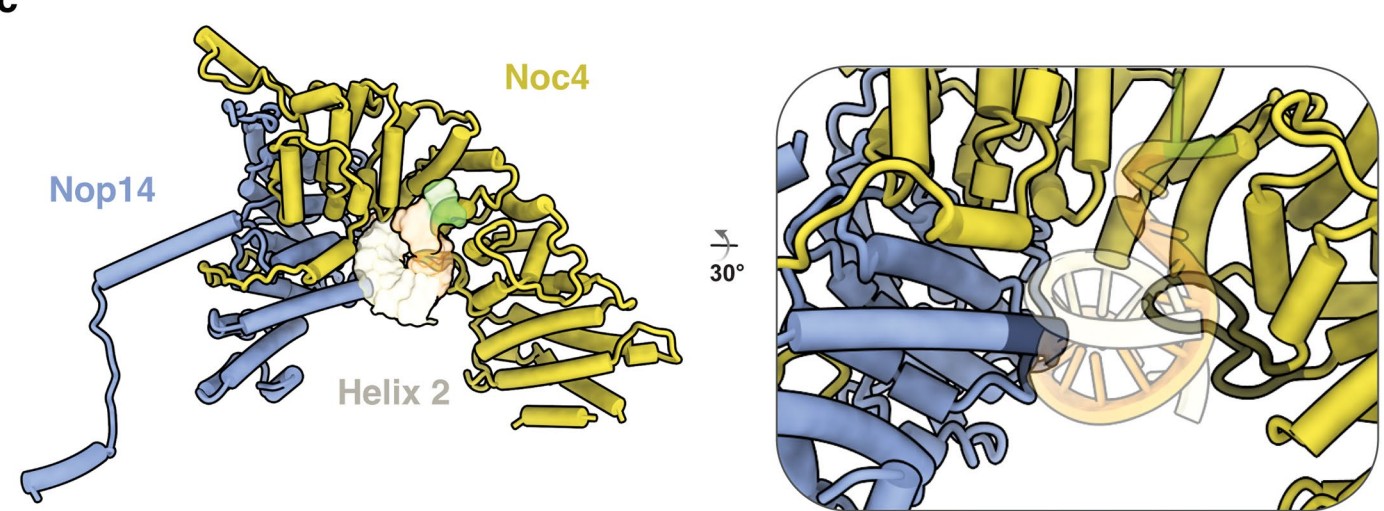

**Extended Data Fig. 7 | Comparison of Noc1-Noc2, Noc2-Noc3 and Noc4-Nop14 heterodimers in yeast.** The evolutionarily related helical repeat heterodimers involved in ribosome biogenesis reveal similar structural composition. **(a)** Noc1-Noc2 heterodimer structure has the unique ability to bind to and chaperone an RNA helix structure (helix 2); zoomed inset shows a rotated view. **(b, c)** In comparison, the yeast Noc2-Noc3 heterodimer and the Noc4-Nop14 heterodimer are both unable to bind RNA in a similar fashion owing to steric hinderance from loops in the central cavity, highlighted with darker color.

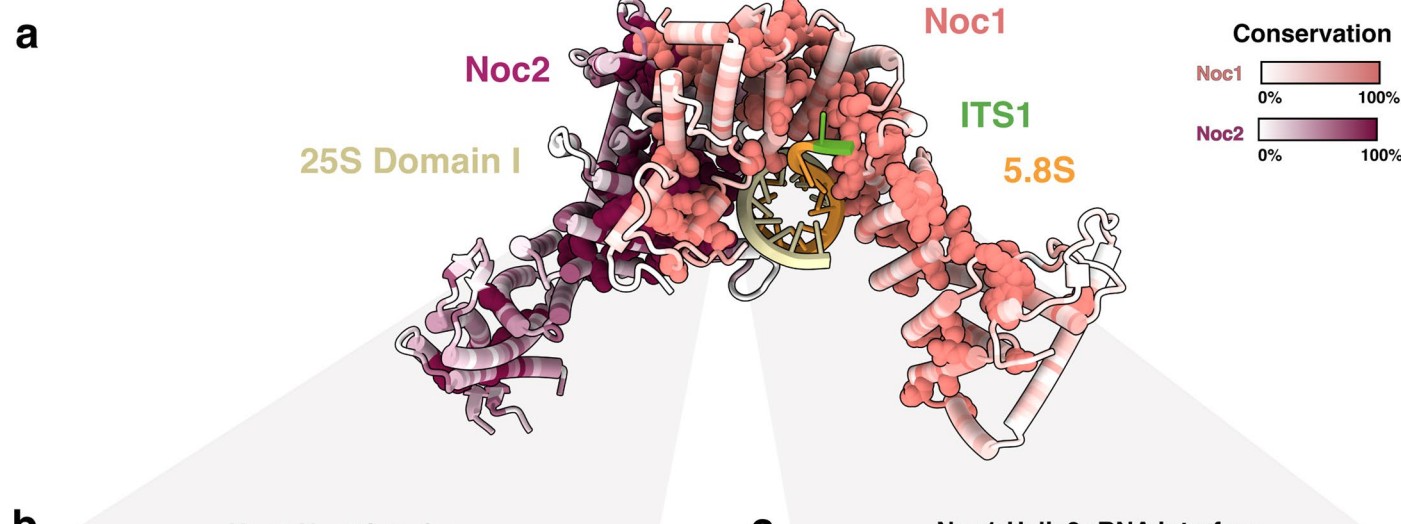

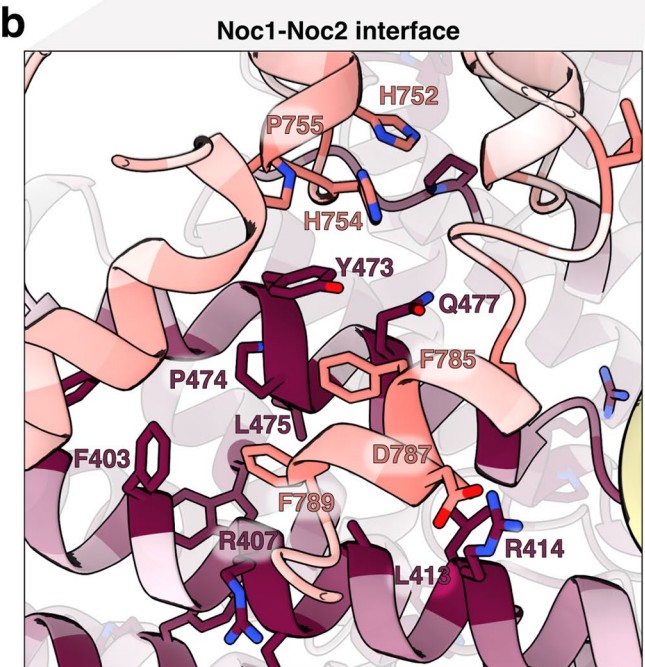

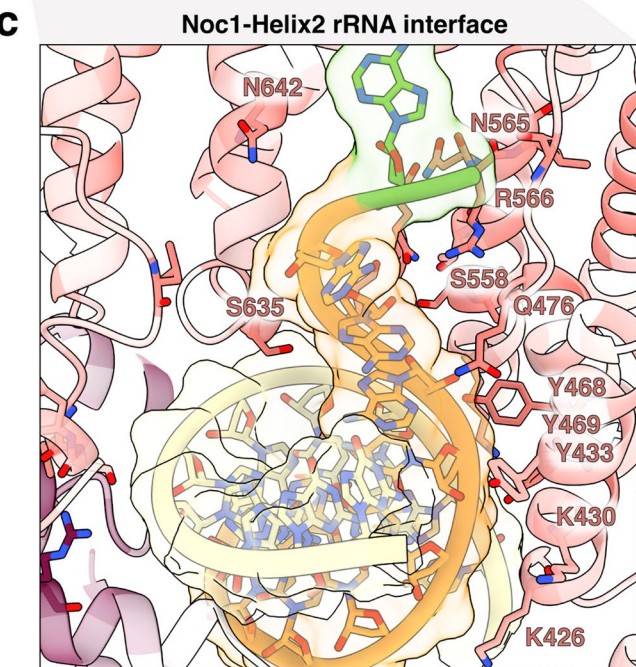

**Extended Data Fig. 8 | Sequence conservation highlights critical interfaces of Noc1 and Noc2. (a)** A top view down the central cavity of the Noc1-Noc2 heterodimer encapsulating helix 2 of the large subunit rRNA. The most conserved residues of the heterodimer are depicted in spheres, with sequence conservation colored from coral (100% conserved) to white (0% conserved) for Noc1, and maroon (100% conserved) to white (0% conserved) for Noc2.

Conservation was calculated by the ConSurf webserver[40], with sequence alignments generated by Clustal Omega[41] from 12 model organisms. **(b)** The Noc1-Noc2 interface highlights clusters of conserved residues that allow for a tight interaction between the two proteins, with conserved residues depicted in sticks. **(c)** The Noc1-RNA interface reveals a set of conserved residues that line the central channel occupied by the RNA of helix 2, depicted in sticks.

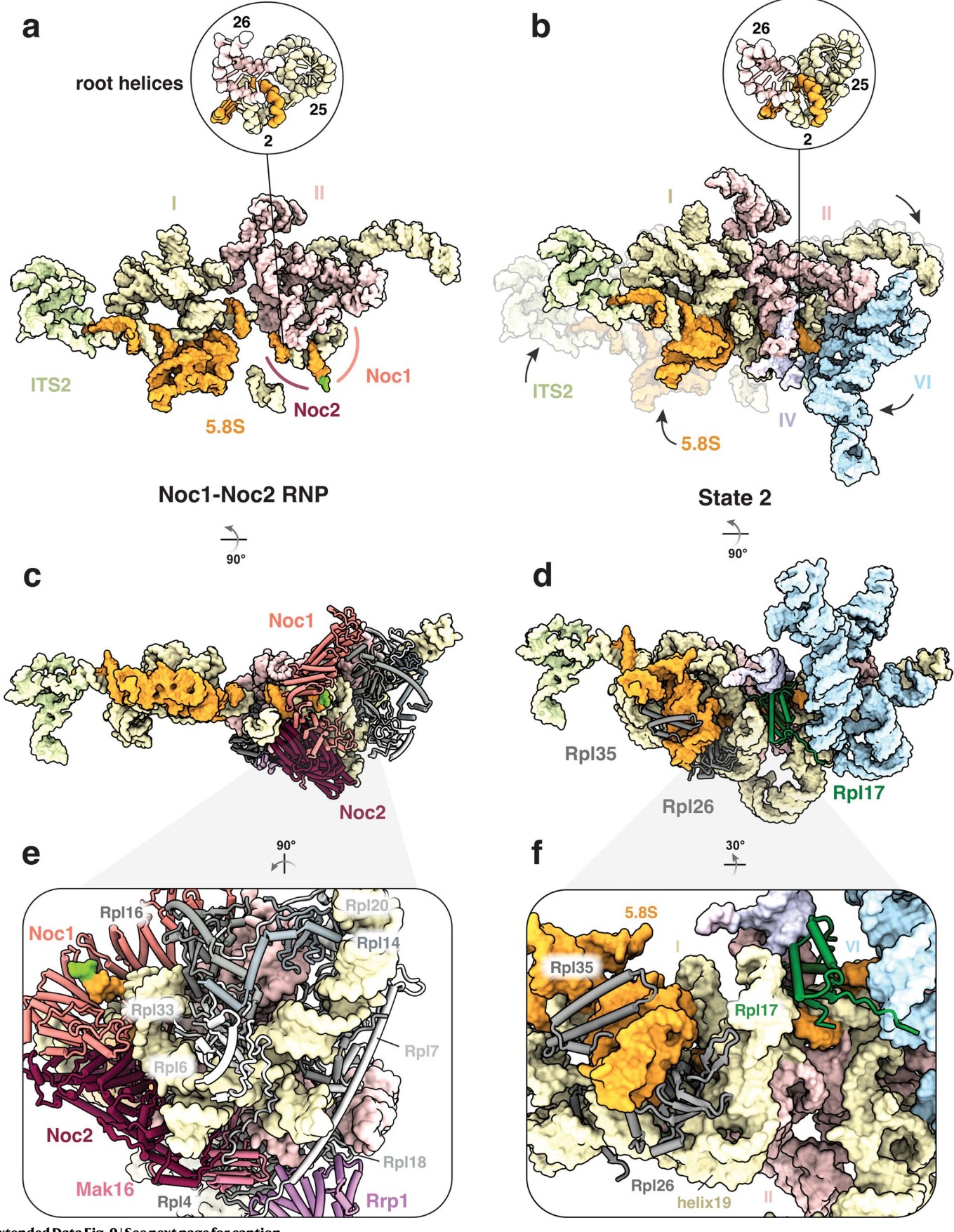

**Extended Data Fig. 9 | See next page for caption.**

**Extended Data Fig. 9 | Analysis of pre-rRNA folding during co- and post-transcriptional ribosome assembly. (a, b)** Comparative view of color-coded rRNA domains within (a) the Noc1-Noc2 RNP and (b) State 2 (PDB 6C0F) with insets highlighting labelled root helices. **(c)** Rotated view with respect to (a). Ribosome assembly factors and ribosomal proteins that stabilize the domain II module are shown as cartoons. **(d)** Rotated view with respect to (b). Ribosomal proteins present at interfaces between rRNA domains are shown as cartoons. **(e)** Zoomed view showing ribosomal proteins stabilizing domains I and II within the Noc1-Noc2 RNP. **(f)** Zoomed view of State 2 (PDB 6C0F) showing ribosomal proteins that stabilize the interface between domains I, II and 5.8S rRNA. Rpl17 is highlighted in green.

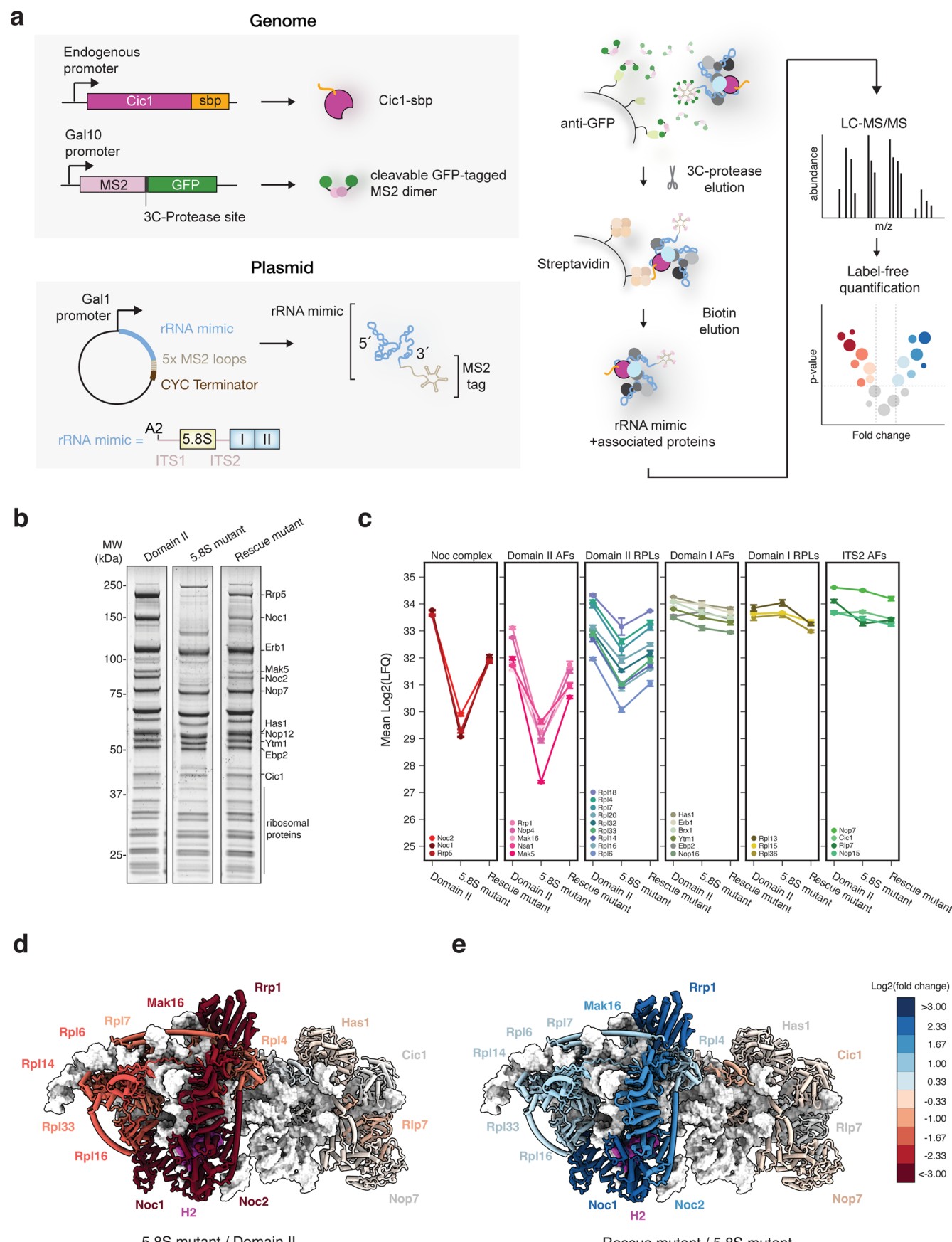

Extended Data Fig. 10 | See next page for caption.

**Extended Data Fig. 10 | Disruption of helix 2 prevents assembly of domain II of the 25S rRNA. (a)** Schematic depiction of two-step purification of Domain II mimics followed by mass-spectrometry analysis. Purification of the mimics has been performed similarly to purification of Noc1 RNP except endogenous Cic1 has been C-terminally tagged with streptavidin-biding peptide (sbp). Following purification, samples were analyzed by liquid chromatography-tandem mass spectrometry (LC-MS/MS) and obtained data normalized using label-free quantification. **(b)** SDS-PAGE analysis of purified Domain II rRNA mimics.

**(c)** Changes in mean protein abundance, expressed as LFQ values, between purified rRNA mimics: Domain II, 5.8S mutant disrupting helix 2, and rescue mutant restoring helix formation. Error bars represent SD of 3 replicates. **(d, e)** Structure of Noc1-Noc2 RNP colored accordingly to log2 fold change in protein abundance between **(d)** 5.8S mutant and Domain II and **(e)** rescue mutant and 5.8S mutant. Proteins are depicted as cartoons and rRNA as white surface. Helix 2 (H2) is shown in magenta.

# Reporting Summary

## Statistics

For all statistical analyses, confirm that the following items are present in the figure legend, table legend, main text, or Methods section.

| n/a | Confirmed | |
|---|---|---|
| ☐ | ☒ | The exact sample size (*n*) for each experimental group/condition, given as a discrete number and unit of measurement |
| ☐ | ☒ | A statement on whether measurements were taken from distinct samples or whether the same sample was measured repeatedly |
| ☐ | ☒ | The statistical test(s) used AND whether they are one- or two-sided<br>*Only common tests should be described solely by name; describe more complex techniques in the Methods section.* |
| ☒ | ☐ | A description of all covariates tested |
| ☐ | ☒ | A description of any assumptions or corrections, such as tests of normality and adjustment for multiple comparisons |
| ☐ | ☒ | A full description of the statistical parameters including central tendency (e.g. means) or other basic estimates (e.g. regression coefficient) AND variation (e.g. standard deviation) or associated estimates of uncertainty (e.g. confidence intervals) |
| ☐ | ☒ | For null hypothesis testing, the test statistic (e.g. *F*, *t*, *r*) with confidence intervals, effect sizes, degrees of freedom and *P* value noted<br>*Give P values as exact values whenever suitable.* |
| ☒ | ☐ | For Bayesian analysis, information on the choice of priors and Markov chain Monte Carlo settings |
| ☒ | ☐ | For hierarchical and complex designs, identification of the appropriate level for tests and full reporting of outcomes |
| ☒ | ☐ | Estimates of effect sizes (e.g. Cohen's *d*, Pearson's *r*), indicating how they were calculated |

*Our web collection on statistics for biologists contains articles on many of the points above.*

## Software and code

Policy information about availability of computer code

| Data collection | Cryo-EM datasets were collected using SerialEM (v. 3.8) on an FEI Titan Krios (FEI/Thermofisher) microscope operating at 300 kV and 22,500x magnification. 8 datasets (Noc1-Noc2 RNP) and 2 datasets (Noc2-Noc3 RNP) were collected at super-resolution pixel size of 0.65 Å/px, with defocus values ranging from -1.0 to -3.0 μm. The total electron dose was 37.9 electrons/Å2 over 32 frames. For comparative IP-MS/MS experiments, samples were analyzed by reversed phase nano-LC-MS/MS using a Fusion Lumos (Thermo Scientific). |
|---|---|
| Data analysis | Cryo-EM movies were aligned and averaged in Relion (v. 3.0) using Relion implementation of MotionCor2-like algorithm, and Contrast transfer function parameters were estimated using CTFFIND4.1. Particles were picked using crYOLO (v. 1.3.5). Relion and cryoSPARC (v. 3.3.1) was used for subsequent classification and refinement steps. Models were manually built in Coot 0.9 using starting models from PDB (code 6C0F) and from the AlphaFold database. Refinement was performed using PHENIX 1.19. Final models were validated using MolProbity, EMRinger, and phenix.rna_validate. For comparative IP-MS/MS experiments, data were quantified and searched against the S. cerevisiae Uniprot protein database (2019) concatenated with the MS2-protein sequence and common contaminations. For the search and quantification, MaxQuant (v. 2.0.3.0) was used. Oxidation of methionine and protein N-terminal acetylation were allowed as variable modifications and all cysteines were treated as being carbamidomethylated. The 'match between runs' option was enabled, and false discovery rates for proteins and peptides were set to 1% and 2% respectively. Protein abundances were expressed as LFQ (label free quantitation) values. Data were analyzed using Perseus (v.1.6.10.50). A two-tailed Student t-test was carried out to confirm statistical significance of data and Q-values resulting from FDR-correction are provided. |

For manuscripts utilizing custom algorithms or software that are central to the research but not yet described in published literature, software must be made available to editors and reviewers. We strongly encourage code deposition in a community repository (e.g. GitHub). See the Nature Portfolio guidelines for submitting code & software for further information.

## Data

Policy information about availability of data

All manuscripts must include a data availability statement. This statement should provide the following information, where applicable:

- Accession codes, unique identifiers, or web links for publicly available datasets
- A description of any restrictions on data availability
- For clinical datasets or third party data, please ensure that the statement adheres to our policy

---

Atomic coordinates and EM maps have been deposited in the Protein Data Bank and Electron Microscopy Data Bank under the following accession codes: Noc1-Noc2 RNP (EMDB-27919, PDB 8E5T) and Noc2-Noc3 RNP (EMD-27910). Raw cryo-EM data has been uploaded to the EMPIAR database (EMPIAR-11379).
Data used but not generated in the study: Starting model for model building of the Noc1-Noc2 RNP (PDB 6C0F).

---

## Human research participants

Policy information about studies involving human research participants and Sex and Gender in Research.

| Reporting on sex and gender | N/A |
|---|---|
| Population characteristics | N/A |
| Recruitment | N/A |
| Ethics oversight | N/A |

Note that full information on the approval of the study protocol must also be provided in the manuscript.

# Field-specific reporting

Please select the one below that is the best fit for your research. If you are not sure, read the appropriate sections before making your selection.

☒ Life sciences ☐ Behavioural & social sciences ☐ Ecological, evolutionary & environmental sciences

For a reference copy of the document with all sections, see nature.com/documents/nr-reporting-summary-flat.pdf

# Life sciences study design

All studies must disclose on these points even when the disclosure is negative.

| Sample size | Sample sizes for cryo-EM datasets were not predetermined. 18,046 (Noc1-Noc2 RNP) and 8,249 (Noc2-Noc3 RNP) micrographs were collected. The number of particles extracted from these micrographs were not predetermined. For comparative IP-MS/MS experiments, three yeast strains were used in technical triplicate for purification and mass spectrometry. |
|---|---|
| Data exclusions | Mis-aligned, damaged, or contaminating particles were excluded from final reconstructions during image processing. For comparative IP-MS/MS experiments, data were filtered such that a protein must be present in all 3 replicates for at least 1 condition. |
| Replication | Immunoprecipitation experiments were repeated independently at least 3 times with similar results. Comparative IP-MS/MS experiments were performed in technical triplicate. Data for comparative IP-MS/MS experiments were gathered from three purification experiments, and three out of three experiments showed similar results. Northern blotting experiments were performed twice independently with similar results. |
| Randomization | During refinement, the gold-standard approach was used to randomly assign particles to half-sets of data that are independently averaged and compared to obtain resolution estimates. Other experiments did not require randomization. |
| Blinding | During single particle analysis, particles are randomly assigned into half-sets, thus no blinding is applicable. No additional grouping was applied. |

# Reporting for specific materials, systems and methods

We require information from authors about some types of materials, experimental systems and methods used in many studies. Here, indicate whether each material, system or method listed is relevant to your study. If you are not sure if a list item applies to your research, read the appropriate section before selecting a response.

## Materials & experimental systems

| n/a | Involved in the study |
|-----|----------------------|
| ☒ | ☐ Antibodies |
| ☒ | ☐ Eukaryotic cell lines |
| ☒ | ☐ Palaeontology and archaeology |
| ☒ | ☐ Animals and other organisms |
| ☒ | ☐ Clinical data |
| ☒ | ☐ Dual use research of concern |

## Methods

| n/a | Involved in the study |
|-----|----------------------|
| ☒ | ☐ ChIP-seq |
| ☒ | ☐ Flow cytometry |
| ☒ | ☐ MRI-based neuroimaging |

