## [Peer Review File · Nature Structural & Molecular Biology]

Peer Review Information

Manuscript Title: A co-transcriptional ribosome assembly checkpoint controls nascent large ribosomal subunit maturation

Corresponding author name(s): Sebastian Klinge

Reviewer Comments & Decisions:

Decision Letter, after successful appeal/first round reviewers

Mess 7th Nov 2022

age:

Dear Dr. Klinge,

Thank you for submission of your manuscript "A co-transcriptional ribosome assembly checkpoint controls nascent large ribosomal subunit maturation". I sincerely apologize for the unusual delay in responding, which is due to our editorial team having been short-staffed in the last months. We have now carefully evaluated the work and discussed it among the editorial team. Unfortunately, we have decided not to consider the manuscript further for publication in Nature Structural & Molecular Biology.

We can only consider a small proportion of the manuscripts submitted to our journal and are often forced to make difficult decisions. Manuscripts are evaluated editorially for their potential interest to a broad audience, the level of novel insight obtained and whether the findings represent a significant advance relative to the published literature, among other considerations.

In this case, we are interested in mechanisms of ribosome biogenesis and appreciate the cryo-EM of the Noc1-Noc2 RNP and the insights it provides into its requirement for the correct folding of helix 2. We recognize that these findings will be of value to others working in this area. However, after discussion among the editorial staff, I am afraid we are not persuaded that the level of functional insight obtained warrants publication in Nature Structural & Molecular Biology.

You might want to consider our sister journal [a href="https://www.nature.com/ncomms/about"](https://www.nature.com/ncomms/about) *Nature Communications* as a potential venue for the publication of these results. *Nature Communications* publishes high quality and influential research and across the full spectrum of the natural sciences. More information on the journal, the potential benefits of transfer and a link to transfer your paper, can be found at the bottom of this email. Please note that the editorial team at *Nature*

Communications will consider your manuscript independently of our suggestion to transfer.

I am sorry we could not be more positive on this occasion. We thank you for the opportunity to consider this work and wish you success in seeking publication elsewhere.

Sincerely,
Sara

Sara Osman, Ph.D.
Associate Editor
Nature Structural & Molecular Biology

Nature Communications is the Nature Portfolio flagship Open Access journal. If you would like this work to be considered for publication there, you can easily transfer the manuscript by following the instructions below. It is not necessary to reformat your paper. Once all files are received, the editors at **Nature Communications** will assess your manuscript's suitability for potential publication; they aim to provide feedback quickly, with a median decision time of 8 days for first editorial decisions on suitability. The journal is also proud to offer double blind and transparent peer review options. For 2019, the 2-year impact factor for **Nature Communications** is 12.121 and the 2-year median is 8 (for further information on journal metrics, please visit our [Nature journals metrics page](http://www.nature.com/npg_/company_info/journal_metrics.html)). Our [open access pages](http://www.nature.com/ncomms/open_access/index.html) contain information about article processing charges, open access funding, and advice and support from Springer Nature.

I suggest that you consider Nature Communications as a suitable venue for your work. To transfer your manuscript there, please use our [Redacted] manuscript transfer portal. You will not have to re-supply manuscript metadata and files, unless you wish to make modifications, but please note that this link can only be used once and remains active until used. For more information, please see our [manuscript transfer FAQ](http://www.nature.com/authors/author_resources/transfer_manuscripts.html?WT.mc_id=EMI_NPG_1511_AUTHORTRANSF&WT.ec_id=AUTHOR) page.

Note that any decision to opt in to In Review at the original journal is not sent to the receiving journal on transfer. You can opt in to [In Review](https://www.nature.com/nature-portfolio/for-authors/in-review) at receiving journals that support this service by choosing to modify your manuscript on transfer. In Review is available for primary research manuscript types only.

For Springer Nature Limited general information and news for authors, see <http://npg.nature.com/authors>

Decision Letter, appeal:**Message:** 5th Jan 2023

Dear Dr. Klinge,

Thank you again for submitting your manuscript "A co-transcriptional ribosome assembly checkpoint controls nascent large ribosomal subunit maturation". We now have comments (below) from the 3 reviewers who evaluated your paper. In light of those reports, we remain interested in your study and would like to see your response to the comments of the referees, in the form of a revised manuscript.

You will see that Reviewers #2 and #3 ask for additional methodological and textual clarifications. Please be sure to address/respond to all concerns of the referees in full in a point-by-point response and highlight all changes in the revised manuscript text file. If you have comments that are intended for editors only, please include those in a separate cover letter.

We expect to see your revised manuscript within 6 weeks. If you cannot send it within this time, please contact us to discuss an extension; we would still consider your revision, provided that no similar work has been accepted for publication at NSMB or published elsewhere.

Reporting Summary:

When submitting the revised version of your manuscript, please pay close attention to our [href="https://www.nature.com/nature-portfolio/editorial-policies/image-integrity">Digital Image Integrity Guidelines.](https://www.nature.com/nature-portfolio/editorial-policies/image-integrity) and to the following points below:

- that unprocessed scans are clearly labelled and match the gels and western blots presented in figures.
- that control panels for gels and western blots are appropriately described as loading on sample processing controls

-- all images in the paper are checked for duplication of panels and for splicing of gel lanes.

Please note that all key data shown in the main figures as cropped gels or blots should be presented in uncropped form, with molecular weight markers. These data can be aggregated into a single supplementary figure item. While these data can be displayed in a relatively informal style, they must refer back to the relevant figures. These data should be submitted with the final revision, as source data, prior to acceptance, but you may want to start putting it together at this point.

Data availability: this journal strongly supports public availability of data. All data used in accepted papers should be available via a public data repository, or alternatively, as Supplementary Information. If data can only be shared on request, please explain why in your Data Availability Statement, and also in the correspondence with your editor. Please note that for some data types, deposition in a public repository is mandatory - more information on our data deposition policies and available repositories can be found below: <https://www.nature.com/nature-research/editorial-policies/reporting-standards#availability-of-data>

[Redacted]

Sincerely,
Sara

Sara Osman, Ph.D.
Associate Editor
Nature Structural & Molecular Biology

Referee expertise:

Referee #1: Cryo-EM, translation

Referee #2: Cryo-EM, ribosome biogenesis

Referee #3: Ribosome biogenesis

Reviewers' Comments:

Reviewer #1:

Remarks to the Author:

The manuscript by Sanghai et al reports the cryo-EM structure of an early assembly precursor of 60S large ribosomal subunit (pre-60S). The structure contains domains I and II and importantly the Noc1-Noc2 module, which has not been found in the many known structures of pre-60S, and thus represents a novel earlier state. The authors also show biochemically that helix 2, which is bound by Noc1-Noc2, is important for pre-60S assembly and A2 site cleavage. The authors have strongly claimed that the assembly of Noc1-Noc2 to helix 2 serves as a quality control check point for pre-60S assembly.

However, this conclusion seems unclear and overstated. The mutational analysis on helix 2 is nice, but there is no evidence whether helix 2 assembly defects occur naturally in cells and are indeed monitored. Overall, the study is technically solid and a nice piece of work that increases our understanding of the early steps of pre-60S assembly. But I feel that the level of advance is not enough to merit publication in NSMB.

Reviewer #2:

Remarks to the Author:

The manuscript by Sanghai et al. presents cryo-EM structures and complementary evidence that early 25S rRNA species are cotranscriptionally stabilized by a complex consisting of Noc1/Noc2. This is a novel and insightful structure, and it is accompanied by some really elegant genetic experiments, including RNA swaps. I believe it will be of sufficient interest to general readers. I recommend publication in Nature Structural and Molecular Biology, provided the authors address a few points below.

Major points:

1. The manuscript would benefit from a much clearer description of the purification and data processing methods, both briefly in the text and in full in the Methods. It is not sufficiently clear on how each purification is done. The Noc1/2 complex is obtained by tagging Noc1 and Noc2. The Noc3 complex whose structure has been described is indicated as critical evidence that the structures obtained represent on-pathway intermediates. The reader should know how that complex was obtained. How does it differ or is it from one of the other purifications. Also, in the Methods, it states something like grids were made from 'multiple purifications.' Which purification went on which grid?
2. The authors should acknowledge the pioneering work of Miller and colleagues by including a citation such as Miller & Beatty, Science 1969.
3. The authors state that the Noc1-Noc2 complex is evolutionarily related to the Noc2-Noc3 complex and others, but do not provide or point to evidence as such, nor state why this is important. Are the "central loops", which the authors suggest inhibit H2 binding (Fig S7 legend), evolutionarily conserved among paralogous proteins?
4. Describe what a 'root helix' is for a general reader.
5. There are no overlays of the EM reconstruction with the corresponding atomic models. This needs to be done. I would recommend in places where the authors are pointing out specific contacts.

Minor point:

1. The authors should describe their statistical analysis in greater detail, e.g. one- or two-tailed t-test and whether or not correction for multiple comparisons is required.

David Taylor

Reviewer #3:

Remarks to the Author:

This is an interesting, high-quality study that reports the purification and structure determination by cryo-EM of an early co-transcriptional pre-60S Noc1-Noc2 RNP complex using a physiologically relevant system in which a pre-rRNA mimic is expressed from a plasmid in vivo in *S. cerevisiae*. The authors also purify a known late pre-60S assembly intermediate (State E) that contains Noc2-Noc3. The significance of the work lies in showing that the Noc1-Noc2 complex controls post transcriptional ribosome assembly at the level of 1) RNA topology, 2) RNA processing, 3) the incorporation of ribosomal proteins, and 4) the chronology of ribosome assembly factor association. A further strength of the work is that the authors functionally test the idea that the Noc1-Noc2 RNP forms a physiologically relevant assembly checkpoint using an elegant combination of yeast genetics, Northern blotting and comparative mass spectrometry. The data are summarized nicely in a model that highlights how the atomic models presented in the current work relate to the historic Miller spread EM images of co-transcriptional ribosome assembly intermediates. Interestingly, the study identifies a role for three Diamond-Blackfan anemia-related proteins in stabilizing the interfaces between domains I, II and the 5.8S rRNA. The links to human ribosomopathy will broaden the appeal of the work to a wide readership beyond the ribosome assembly community. The manuscript is well written with informative, clear figures. A wealth of supporting technical detail is provided in the extensive supplementary methods and figures that appropriately discuss the cryo-EM data collection, processing and classification strategies used and robustly support the conclusions that the authors wish to draw.

I have only a few minor comments:

1. All the work presented in the study is performed in yeast. Presumably the proposed checkpoint is likely to be conserved in mammalian cells, but are there any data to support this? Perhaps consider adding a statement in the text to address this point or add "in yeast" to the title? Interestingly, a very recent study (<https://doi.org/10.1242/jcs.260110>) shows that Noc1 is critical for pre-rRNA processing, growth and cell competition in *Drosophila* and would be worth citing.
2. I find some of the same-color labels difficult to read eg Extended data Fig 8. Consider changing to black font.
3. Some of the proteins identified in the mass spec analysis in Extended Data Fig. 1e are not seen in the final reconstruction (eg Rex4, Nop12, Nop4, Nop16). Please comment.
4. RP nomenclature is confusing and inconsistent. Why are some of the yeast ribosomal proteins shown in upper case, but assembly factors in lower case in the Figures? Supp Table 2 uses lower case and the new nomenclature. Extended Fig 9 has Rpl17 in the legend, but upper case RPL17 in the Figure. On p7, the authors refer to DBA ribosomal proteins RPL17, RPL26 and RPL35A (but RPL35A is the human nomenclature?). Do they mean uL33 using the new nomenclature? Please unify and rationalize the nomenclature throughout the text.
5. The inclusion of Rpl17/RPL17 as a DBA protein appears to be based on the identification of variants in 2 members of a single Swiss family reported in the cited reference 23. It is important to note that although supported by functional analysis in zebrafish, the

pathogenicity of this variant and therefore its formal assignment as a DBA protein, needs further validation.

5. Please include the original reference for the Miller hypotonic lysis chromatin spreading method:
Miller, O.L. Jr. and Beatty, B.R. (1969) Visualization of nucleolar genes. Science 164, 955-957

Author Rebuttal, first revision:

We thank the three reviewers for their helpful comments, which we address below.

Referee expertise:

Referee #1: Cryo-EM, translation

Referee #2: Cryo-EM, ribosome biogenesis

Referee #3: Ribosome biogenesis

Reviewers' Comments:

Reviewer #1:

Remarks to the Author:

The manuscript by Sanghai et al reports the cryo-EM structure of an early assembly precursor of 60S large ribosomal subunit (pre-60S). The structure contains domains I and II and importantly the Noc1-Noc2 module, which has not been found in the many known structures of pre-60S, and thus represents a novel earlier state. The authors also show biochemically that helix 2, which is bound by Noc1-Noc2, is important for pre-60S assembly and A2 site cleavage. The authors have strongly claimed that the assembly of Noc1-Noc2 to helix 2 serves as a quality control check point for pre-60S assembly. However, this conclusion seems unclear and overstated. The mutational analysis on helix 2 is nice, but there is no evidence whether helix 2 assembly defects occur naturally in cells and are indeed monitored. Overall, the study is technically solid and a nice piece of work that increases our understanding of the early steps of pre-60S assembly.

We thank reviewer 1 for the appreciating the impact of our work.

But I feel that the level of advance is not enough to merit publication in NSMB.

Here we respectfully disagree and refer to the assessment of reviewers 2 and 3.

Reviewer #2:

Remarks to the Author:

The manuscript by Sanghai et al. presents cryo-EM structures and complementary evidence that early 25S rRNA species are cotranscriptionally stabilized by a complex consisting of Noc1/Noc2. This is a novel and insightful structure, and it is accompanied by some really elegant genetic experiments, including RNA swaps. I believe it will be of sufficient interest to general readers. I recommend publication in Nature Structural and Molecular Biology, provided the authors address a few points below.

We thank reviewer 2 for the positive assessment of our work and have addressed the individual points below.

Major points:

1. The manuscript would benefit from a much clearer description of the purification and data processing methods, both briefly in the text and in full in the Methods. It is not sufficiently clear on how each purification is done. The Noc1/2 complex is obtained by tagging Noc1 and Noc2. The Noc3 complex whose structure has been described is indicated as critical evidence that the structures obtained represent on-pathway intermediates. The reader should know how that complex was obtained. How does it differ or is it from one of the other purifications.

We agree with reviewer 2 that the purification of these complexes should be described in more detail in the text and have therefore clarified the text on page 4 as follows:

“By using the pre-rRNA mimic with MS2 RNA aptamers and tagged Noc1 as baits for affinity purification, we were able to isolate an early co-transcriptional assembly intermediate (hereafter referred to as the Noc1-Noc2 RNP) (Extended Data Fig. 1). Similarly, by using the same prerRNA mimic with MS2 RNA aptamers and tagged Noc2 we were able to isolate the late nucleolar pre-60S assembly intermediate (State E, containing Noc2-Noc3)¹⁶, indicating our pre-rRNA mimic proceeds along a physiologically relevant assembly pathway (Extended Data Fig. 2).”

In addition to the indicated changes in the main text, we have expanded the methods section with the following sentences on page 23:

“The resulting strains were used for the purification of the Noc1-Noc2 RNP (MS2-3C-GFP; Noc1-sbp), State E (MS2-3C-GFP; Noc2-sbp) and pre-60S particles for comparative mass spectrometry (MS2-3C-GFP; Cic1-sbp). These strains were subsequently transformed with the pESC_URA plasmids coding for pre-60S rRNA mimics for transient expression.”

Also, in the Methods, it states something like grids were made from 'multiple purifications.' Which purification went on which grid?

To further clarify how the eight separate cryo-EM datasets were obtained for the Noc1Noc2 RNP, we expanded the methods section on page 25 with the following statement:

“Eight separate purifications of the Noc1-Noc2 RNP were performed to prepare eight separate cryo-EM grids that were used to collect eight data sets (DS1-DS8).”

2. The authors should acknowledge the pioneering work of Miller and colleagues by including a citation such as Miller & Beatty, Science 1969.

We thank reviewers 2 and 3 for spotting this oversight and have introduced this reference (new reference 2).

3. The authors state that the Noc1-Noc2 complex is evolutionarily related to the Noc2-Noc3 complex and others, but do not provide or point to evidence as such, nor state why this is important. Are the “central loops”, which the authors suggest inhibit H2 binding (Fig S7 legend), evolutionarily conserved among paralogous proteins?

The Noc1-Noc2, Noc2-Noc3 and Nop14-Noc4 complexes adopt similar shapes, and have previously been shown to be evolutionarily related, which we have now referenced on page 6 (reference 22). The loops displayed in Extended Data Fig. 7 only show the available structural data in yeast and we are not aware of a conservation of these elements. To clarify this further, we have edited the figure legend title to the following:

“Extended Data Fig. 7. Comparison of Noc1-Noc2, Noc2-Noc3 and Noc4-Nop14 heterodimers in yeast.”

4. Describe what a 'root helix' is for a general reader.

We thank reviewer 2 for this point and have added the definition at the earliest possible instance, on page 5:

“Since root helices form the base of each of the 25S rRNA subdomains (I-VI) and hence the architectural core of the LSU^{20,21}, their early recognition by the Noc1-Noc2 complex indicates a key point of quality control.”

5. There are no overlays of the EM reconstruction with the corresponding atomic models. This needs to be done. I would recommend in places where the authors are pointing out specific contacts.

We agree with reviewer 2 that this is important and in addition to detailed views of atomic models and density maps which we previously showed in Extended Data Fig. 6, we have now also included a view that highlights the essence of the structural data, namely the recognition of root helices by Noc1-Noc2 in Extended Data Fig. 6a. Given the intermediate resolution of our composite map ~4 Ångstroms, we have been cautious in our

interpretation and avoided drawing conclusions about interactions at the side-chain level in the manuscript.

Minor point:

1. The authors should describe their statistical analysis in greater detail, e.g. one- or two-tailed ttest and whether or not correction for multiple comparisons is required.

David Taylor

We agree with reviewer 2 that this should be more clearly explained and now mention that a two-tailed student t-test was carried out.

As outlined in the methods (page 30), an FDR correction was applied for proteins and peptides at 1% and 2% respectively. We further note that in the Volcano plots shown in Figs. 3f,g only proteins with p-values of 10^{-4} are displayed. We are therefore highlighting only proteins with a cut-off that is two orders of magnitude more stringent than the typical p-value of 0.01. In addition, calculated Q-values, which are a result of FDR-correction are listed in the supplementary mass spectrometry data.

Reviewer #3:

Remarks to the Author:

This is an interesting, high-quality study that reports the purification and structure determination by cryo-EM of an early co-transcriptional pre-60S Noc1-Noc2 RNP complex using a physiologically relevant system in which a pre-rRNA mimic is expressed from a plasmid in vivo in *S. cerevisiae*. The authors also purify a known late pre-60S assembly intermediate (State E) that contains Noc2-Noc3. The significance of the work lies in showing that the Noc1-Noc2 complex controls post transcriptional ribosome assembly at the level of 1) RNA topology, 2) RNA

processing, 3) the incorporation of ribosomal proteins, and 4) the chronology of ribosome assembly factor association. A further strength of the work is that the authors functionally test the idea that the Noc1-Noc2 RNP forms a physiologically relevant assembly checkpoint using an elegant combination of yeast genetics, Northern blotting and comparative mass spectrometry. The data are summarized nicely in a model that highlights how the atomic models presented in the current work relate to the historic Miller spread EM images of co-transcriptional ribosome assembly intermediates. Interestingly, the study identifies a role for three Diamond-Blackfan anemia-related proteins in stabilizing the interfaces between domains I, II and the 5.8S rRNA. The links to human ribosomopathy will broaden the appeal of the work to a wide readership beyond the ribosome assembly community. The manuscript is well written with informative, clear figures. A wealth of supporting technical detail is provided in the extensive supplementary methods and figures that appropriately discuss the cryo-EM data collection, processing and classification strategies used and robustly support the conclusions that the authors wish to draw.

We thank reviewer 3 for the positive assessment of our work.

I have only a few minor comments:

1. All the work presented in the study is performed in yeast. Presumably the proposed checkpoint is likely to be conserved in mammalian cells, but are there any data to support this? Perhaps consider adding a statement in the text to address this point or add “in yeast” to the title? Interestingly, a very recent study (<https://doi.org/10.1242/jcs.260110>) shows that Noc1 is critical for pre-rRNA processing, growth and cell competition in *Drosophila* and would be worth citing.

We agree with reviewer 3 that all our data is from *Saccharomyces cerevisiae*. However, given the highly conserved nature of early eukaryotic ribosome assembly, we believe that the observed principles are generally applicable to other species and have edited the conclusion of our manuscript to include the following statement, which includes the suggested reference (here reference 6):

“Our structural and functional data in yeast shed light on the earliest events during cotranscriptional large ribosomal subunit formation, which are highly conserved in eukaryotes²⁶.”

2. I find some of the same-color labels difficult to read eg Extended data Fig 8. Consider changing to black font.

To address this point, we have increased the font size and added a black outline so that the color coding of the labels remains.

3. Some of the proteins identified in the mass spec analysis in Extended Data Fig. 1e are not seen in the final reconstruction (eg Rex4, Nop12, Nop4, Nop16). Please comment.

As observed for many ribosome assembly intermediates, cryo-EM reconstructions only capture the most ordered regions of a given particle. In our case, relative to the two wellordered domains I and II, we believe that several proteins (such as Rex4, Nop12, Nop4 and Nop16) and RNA segments are present as confirmed by mass spectrometry but not visualized by cryo-EM.

4. RP nomenclature is confusing and inconsistent. Why are the some of the yeast ribosomal proteins shown in upper case, but assembly factors in lower case in the Figures? Supp Table 2 uses lower case and the new nomenclature. Extended Fig 9 has Rpl17 in the legend, but upper case RPL17 in the Figure. On p7, the authors refer to DBA ribosomal proteins RPL17, RPL26 and RPL35A (but RPL35A is the human nomenclature?). Do they mean uL33 using the new nomenclature? Please unify and rationalize the nomenclature throughout the text.

We thank reviewer 3 for spotting this and have therefore used the yeast nomenclature throughout the manuscript. The label RPL35A has now been corrected to Rpl35 (yeast nomenclature; uL29 universal nomenclature).

5. The inclusion of Rpl17/RPL17 as a DBA protein appears to be based on the identification of variants in 2 members of a single Swiss family reported in the cited reference 23. It is important to note that although supported by functional analysis in zebrafish, the pathogenicity of this variant and therefore its formal assignment as a DBA protein, needs further validation.

We agree that a formal assignment as a DBA protein is still missing for Rpl17/RPL17 and have amended the relevant sentence as follows:

“The importance of critical interfaces between domains I, II and the 5.8S rRNA is further highlighted by the finding that key stabilizers of these sites include Rpl17(RPL17) and DiamondBlackfan anemia proteins Rpl26(RPL26) and Rpl33A(RPL35A) (Extended Data Fig. 9c-f)^{22,23}.”

5. Please include the original reference for the Miller hypotonic lysis chromatin spreading method: Miller, O.L. Jr. and Beatty, B.R. (1969) Visualization of nucleolar genes. Science 164, 955-957

We thank reviewers 2 and 3 for spotting this oversight and have introduced this reference (new reference 2).

Decision Letter, after first round of revisions

Message: Our ref: NSMB-A46944B

24th Jan 2023

Dear Dr. Klinge,

Thank you for submitting your revised manuscript "A co-transcriptional ribosome assembly checkpoint controls nascent large ribosomal subunit maturation" (NSMB-A46944B). It has now been assessed editorially, and we find the paper has improved in revision, and therefore we'll be happy in principle to publish it in Nature Structural & Molecular Biology, pending minor revisions to comply with our editorial and formatting guidelines.

We are now performing detailed checks on your paper and will send you a checklist detailing our editorial and formatting requirements in the next couple of weeks. Please do not upload the final materials and make any revisions until you receive this additional information from us.

To facilitate our work at this stage, we would appreciate if you could send us the main text as a word file. Please make sure to copy the NSMB account (cc'ed above).

Sincerely,
Sara

Sara Osman, Ph.D.
Associate Editor
Nature Structural & Molecular Biology

Final Decision Letter:

Message 24th Feb 2023

:

Dear Dr. Klinge,

We are now happy to accept your revised paper "A co-transcriptional ribosome assembly checkpoint controls nascent large ribosomal subunit maturation" for publication as a Article in Nature Structural & Molecular Biology.

Your paper will be published online soon after we receive proof corrections and will appear in print in the next available issue. You can find out your date of online publication by contacting the production team shortly after sending your proof corrections. Content is published online weekly on Mondays and Thursdays, and the embargo is set at 16:00 London time (GMT)/11:00 am US Eastern time (EST) on the day of publication. Now is the time to inform your Public Relations or Press Office about your paper, as they might be interested in promoting its publication. This will allow them time to prepare an accurate and satisfactory press release. Include your manuscript tracking number (NSMB-A46944C) and our journal name, which they will need when they contact our press office.

About one week before your paper is published online, we shall be distributing a press release to news organizations worldwide, which may very well include details of your work. We are happy for your institution or funding agency to prepare its own press release, but it must mention the embargo date and Nature Structural & Molecular Biology. If you or your Press Office have any enquiries in the meantime, please contact press@nature.com.

You can now use a single sign-on for all your accounts, view the status of all your manuscript submissions and reviews, access usage statistics for your published articles and

download a record of your refereeing activity for the Nature journals.

Please note that *Nature Structural & Molecular Biology* is a Transformative Journal (TJ). Authors may publish their research with us through the traditional subscription access route or make their paper immediately open access through payment of an article-processing charge (APC). Authors will not be required to make a final decision about access to their article until it has been accepted. <https://www.springernature.com/gp/open-research/transformative-journals> Find out more about Transformative Journals

Sincerely,

Dimitris Typas
Associate Editor
Nature Structural & Molecular Biology
ORCID: 0000-0002-8737-1319
